# Controls of aeolian and fluvial sediment influx to the northern Red Sea over the last 220 000 years

Werner Ehrmann[1], Paul A. Wilson[2], Helge W. Arz[3], Gerhard Schmiedl[4]

[1] Institut für Erdsystemwissenschaft und Fernerkundung, Universität Leipzig, Talstraße 35, 04103 Leipzig, Germany
[2] University of Southampton, Waterfront Campus, National Oceanography Centre, Southampton, SO14 3ZH, United Kingdom
[3] Leibniz-Institut für Ostseeforschung, Seestraße 15, 18119 Warnemünde, Germany
[4] Institut für Geologie, Centrum für Erdsystemforschung und Nachhaltigkeit, Universität Hamburg, Bundesstraße 55, 20146 Hamburg, Germany

*Correspondence to*: Werner Ehrmann (ehrmann@uni-leipzig.de)

**Abstract.** Present-day sediment influx to the northern Red Sea is dominated by aeolian dust because of its position between the large deserts of northern Africa, the Arabian Peninsula and the Levant, and the absence of discharge
from perennial rivers. However, sediment cores retrieved from the northern Red Sea reveal strong temporal variability of dust influx to the basin on glacial-interglacial timescales and several shorter-term strong episodes of fluvial input. We report new palaeoclimate and sediment provenance records for the last ca. 220 kyr from marine sediment core KL23, retrieved from the northern part of this basin. Our data suggest that the Nile delta became a major dust source during glacial conditions, in response to the glacioeustatic sea-level fall and associated subaerial
exposure of volcanic-rich debris originally transported down the river Nile from the Ethiopian Highlands. Windblown dust from this delta source is characterized by high smectite concentrations and Ti contents. It is transported to the northern Red Sea on prevailing NNW winds. Our data also suggest a contribution of kaolinite-rich windblown dust from Egypt, Sinai and the Levant to KL23 on the same winds. The activity of this source is hydrologically controlled, with minima in kaolinite concentrations documenting phases of increased humidity,
probably due to enhanced Mediterranean cyclogenesis and a southward expansion of the Mediterranean winter rains. Short-term reactivations of wadi systems during fluvial episodes are identified by maxima in the abundance of clay-sized terrigenous sediment components, high chlorite concentrations and high $\varepsilon_{Nd}$. These episodes correlate with phases of reduced aeolian influx to the northern Red Sea and coincide with African Humid Periods, both in timing and relative intensity. This result implies that the Mediterranean climate system and the African monsoon
are closely coupled.

## 1 Introduction

The Red Sea is a NNW–SSE oriented, ca. 2200 km long and at most ca. 350 km wide deep-sea basin situated between northern Africa and the Arabian Peninsula. Its only connection to the open ocean is via the narrow and shallow strait of Bab al-Mandab in the south (Fig. 1a). Two major climate and wind systems affect the Red Sea, east Mediterranean cyclones (Cyprus cyclones) in its northernmost part, and the African and Arabian monsoon system in its central and southern parts. North of 19°N, winds blow dominantly from the NNW throughout the year, south of 19°N, the wind system is characterized by seasonal changes of the African and Arabian monsoon system, with prevailing NNW summer winds and stronger SSE winter winds. Additionally, eastward blowing winds originate from the Tokar Gap at ca. 18°N on the African continent and spread towards Saudi Arabia, and westward jets blow through narrow valleys in the northern part of the Arabian Peninsula (Edwards, 1987; Jiang et al., 2009; Langodan et al., 2014; Enzel et al., 2015; Menezes et al., 2018).

Under modern (semi)arid climate conditions, no perennial rivers actively discharge into the Red Sea. Instead, terrigenous sediment influx to the basin is dominated by mineral dust, because the basin is surrounded by dust-active regions in the large deserts of northern Africa, the Horn of Africa, the Arabian Peninsula, the Levant, and Mesopotamia (Tegen et al., 2002; Engelstaedter et al., 2006; Stuut et al., 2009; Schepanski et al., 2012; Notaro et al., 2013; Scheuvens et al., 2013; Ramaswamy et al., 2017; Palchan and Torfstein, 2019; Kunkelova et al., 2022, 2024). Geological records, however, indicate profound hydrological change in these source regions over the last ~220 thousand years (kyr), repeatedly leading to intervals of reduced dust influx and activation of wadis with fluvial sediment supply to the Red Sea.

In the central Red Sea, past hydrological change was controlled by the intensity of the tropical rainfall and vegetation cover driven by insolation-controlled African monsoon circulation, the local expression of the African Humid Periods (AHPs; Rossignol-Strick, 1983; Arz et al., 2003; Stein et al., 2007; Palchan et al., 2013; Hartman et al., 2020; Duque-Villegas et al., 2024; Ehrmann et al., 2024). The northern Red Sea is less well studied but the records available suggest changes in dust supply paced mainly by glacial/interglacial cycles (not local insolation) with glacials characterized by arid conditions and strong dust influx (Palchan et al., 2013; Hartman et al., 2020). This result merits further attention because it suggests that the northern Red Sea may archive information on atmospheric teleconnections between the high northern latitudes, where the warming response to human-driven change is fastest, and a region that is already water-stressed and widely projected to dry in the coming decades (e.g., Seager et al., 2014).

Here we report new palaeoclimate records for the last ca. 220 kyr from marine core KL23, which was retrieved from the northern Red Sea (Fig. 1). We use high-resolution clay mineral, geochemical, and grain size data together with Nd and Sr isotope data to fingerprint sediment provenance and to reconstruct changes in the intensity of aeolian and fluvial sediment influx through time. The surrounding desert landscape and the complex prevailing wind systems can make provenance-determination challenging in the region. We, therefore, exploit recent advances in quantifying modern dust source activation

frequency and geochemical fingerprinting of different sources in ocean sediment cores (e.g., Jewell et al., 2021, 2022; Crocker et al., 2022; Kunkelova et al., 2022, 2024).

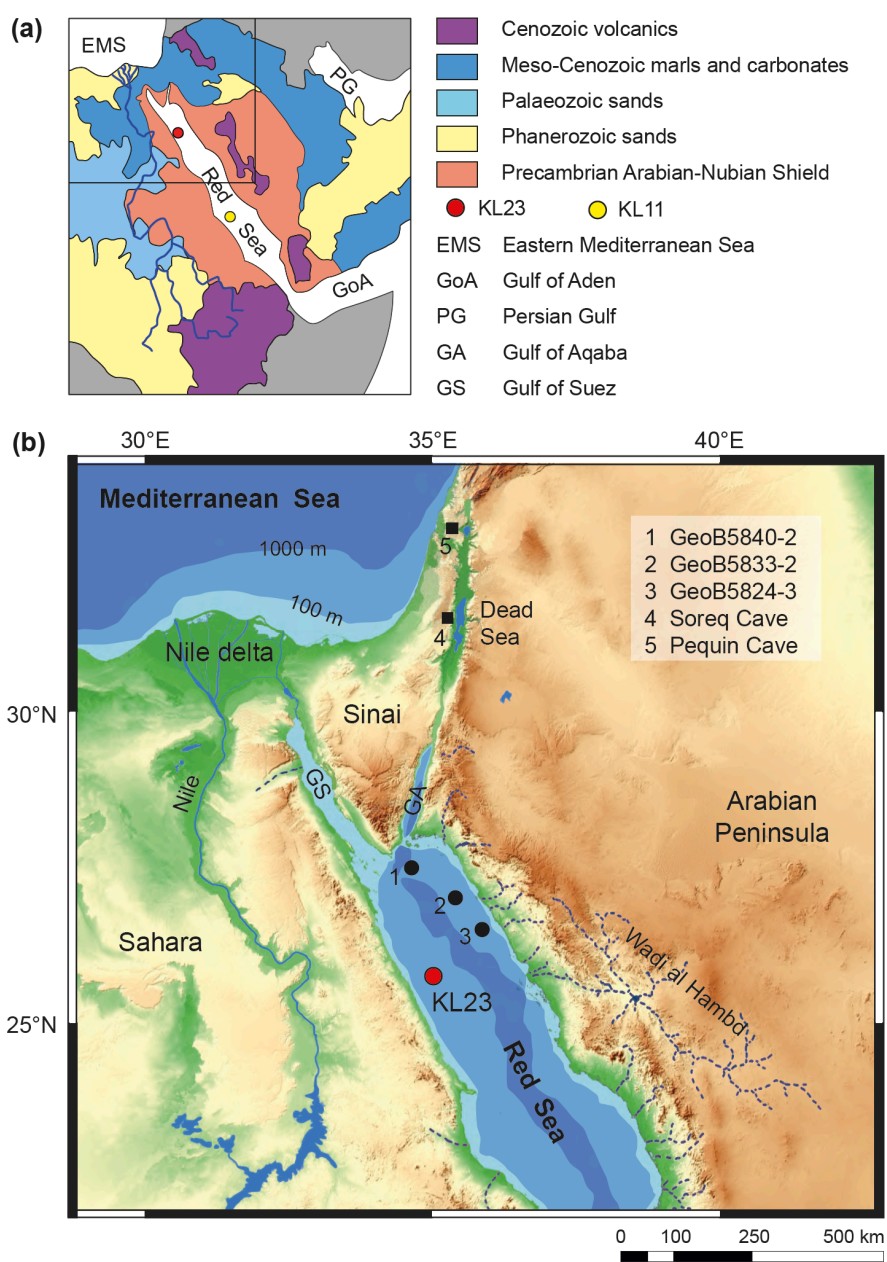

**Figure 1. Location of the investigated marine sediment core KL23 in the northern Red Sea and other records mentioned**
**in the text. (a) Simplified geology of the hinterland of the Red Sea (after Stein et al., 2007, and Palchan et al., 2013). The box at the top left indicates the region shown in panel (b). (b) Hypsometric map of the area around the northern Red Sea (source: OpenTopoMap; CC-BY-SA).**

## 2 Material and Methods

Sediment core KL23 was retrieved in 1995 during RV *Meteor* expedition M31/2 from the northern Red Sea at 25°44.9′ N and
35°03.3′ E; the water depth was 702 m (Hemleben et al., 1996; Badawi et al., 2005). We investigated the upper 12.60 m of the
22.10 m long core. The KL23 sediments are dominantly grey to olive grey and olive in colour. They consist mainly of
calcareous ooze. No tephra layers were detected. The core seems to archive a continuous sediment succession, since no
indications of erosion, slumping, debris flows or turbidites were detected.

### 2.1 Age model

The age model for the upper 920 cm of KL23 was taken from Hartman et al. (2020), who correlated the oxygen isotope record
of KL23 with the age model of KL11. The latter is based on a comparison of the planktic foraminiferal stable oxygen isotope
record with the U–Th dated speleothem record of Soreq cave (Grant et al., 2012) and integrated eight accelerator mass
spectrometry (AMS) radiocarbon ages (Hartman et al., 2020). The age model for the interval 920–1260 cm of KL23 was
constructed by tuning the benthic $\delta^{18}$O record (Geiselhart, 1998) to the sea level curve for the Red Sea (Grant et al., 2014),
since the $\delta^{18}$O signal of Red Sea deep water is strongly dominated by sea level changes (Siddall et al., 2003). Graphical tie
points comprise 1050 cm (170.7 ka), 1167 cm (197.8 ka), and 1271 cm (226.6 ka). Figure S1 in the supplement provides an
age / depth plot and bulk sedimentation rates of core KL23.

### 2.2 Clay mineralogy, grain size and sediment geochemistry

We analysed ca. 500 samples from KL23 for clay mineralogy, grain size, and carbonate content. The terrigenous components
were isolated from the bulk sediment by removing carbonate and organic matter with 10 % acetic acid and 5 % hydrogen
peroxide, respectively. Because remains of siliceous microfossils were detected only sporadically in the sediments, the content
of terrigenous matter could be calculated by weight difference.

The investigation of the mineralogical composition of the clay fraction (<2 µm) by X-ray diffraction (XRD) followed standard
procedures (Ehrmann and Schmiedl, 2021). The grain size analyses of the carbonate-free sediment fraction were performed
using a laser particle sizer (Analysette 22, Fritsch GmbH). See supplement for a more comprehensive description of the
methods.

We measured major and trace elements of KL23 at the Leibniz-Institut für Ostseeforschung, Warnemünde, Germany, with an
ITRAX (Cox Analytical Systems) X-ray fluorescence (XRF) core scanner. See supplement for more details. The elements Si,
K, Ti, Al, Rb and Zr are assumed to describe the basic chemical composition of the terrigenous sediment fraction, because
they are common components in weathering products of granitoid, metamorphic and volcanic rocks and clastic sediments, but

not in biogenic components (e.g., Croudace and Rothwell, 2015). They are reported normalised to total terrigenous element counts by calculating ratios of element counts / total terrigenous element counts (e.g., Ti / terr). In addition, we calculated the ratio of total terrigenous counts to Ca counts (Terr / Ca).

## 2.3 Nd and Sr isotopes

Using the high-resolution downcore XRF and clay mineral data sets we selected representative samples for Sr and Nd isotope analysis of the bulk terrigenous fraction. Our analytical programme followed the method of Jewell et al. (2022) which includes treatment to remove marine mineral phases that, if not removed from ocean sediment cores, can seriously contaminate the geochemical fingerprint of the terrigenous fraction. For further details see Jewell et al. (2022). These data were compared to the geochemical fingerprint of continental sediment sources following Crocker et al. (2022) and Ehrmann et al. (2024). Briefly, downcore records are compared to the geochemical fingerprint of continental sources emphasising the approach of Jewell et al. (2021) and Kunkelova et al. (2022) wherein the composition of unconsolidated sediment samples is weighted by dust source activation frequency (DSAF) (e.g., Schepanski et al., 2007). See Kunkelova et al. (2024) for a composite DSAF map covering all of North Africa, the Horn of Africa and westernmost Asia. See supplement for a more comprehensive description of the method.

## 2.4 Spectral analyses

Spectral analyses were based on the REDFIT algorithm (Schulz and Mudelsee, 2002) as implemented in the PAST software, version 4.11 (Hammer et al., 2001). A Welch window, an oversampling factor of 10, and two segments with overlapping by 50 % were applied. "False-alarm" was based on parametric approximation (chi2).

## 3 Results

Results of our investigations of Nd and Sr isotopes are presented in Figs. 2 and 6. The radiogenic isotope data on the cleaned bulk terrigenous fraction in samples from KL23 fall in the following ranges: $\varepsilon_{Nd}$: –7.16 to –2.39 and $^{87}Sr$ / $^{86}Sr$: 0.71406. to 0.70990.

Results of clay mineral analyses are presented in Figs. 3, 5, 6, and S2. Smectite is the dominant clay mineral in the KL23 sediments. Its concentrations of about 25 %–60 % are strongly linked to downcore glacial/interglacial-paced cycles, with high concentrations occurring in the glacial intervals. Illite and palygorskite show concentrations of 10 %–20 % and 5 %–10 %, respectively, and show a rough negative correlation to the smectite concentrations. Chlorite shows relatively low and constant background concentrations of 5 %–10 %, but distinct maxima of up to ca. 25 % are observed at 219, 198, 128, 111, 107, 85,

and 11 ka. Kaolinite concentrations generally increase between 224 ka and 100 ka from 25 % to 35 %, then decrease to ca. 20 %–25 % during the Last Glacial Maximum, before increasing again to ca. 35 %. Distinct minima occur at ca. 219, 198 and 128 ka. As a result of the closed sum effect, a reduction in the concentration of one clay mineral may be caused by an increase in the concentrations of other clay minerals. Smectite as the dominant clay mineral shows the largest amplitude in its
concentration pattern. We consider it unlikely that the documented changes in smectite content were driven by dilution, because this explanation would require that the other four less abundant clay minerals fluctuated together with one another in the same direction. Thus, we infer that smectite variability is the primary pacemaker of change in the clay mineral assemblage.

The concentration of terrigenous components in KL23 sediments fluctuates between ca. 20 % and 50 %, with maxima
occurring during glacial periods and minima during interglacial periods (Fig. 3e). The mean content of terrigenous matter is ca. 37 %. Thus, some 63 % of the sediment consist of biogenic calcareous components.

Grain size data are presented in Figs. 6 and S3. Particles of the silt grain size (2 µm–63 µm) dominate the terrigenous sediment fraction of KL23. Highest concentrations up to 95 % occur during the glacial periods, lowest concentrations of ca. 65 % in
Marine Isotope Stage (MIS) 1. Concentrations of the clay fraction (<2 µm) fluctuate between 3 % and 20 %, with lowest concentrations during the last and the penultimate glacial maxima and highest values at ca. 198 ka and 128 ka. Typically, sand (>63 µm) occurs only in trace amounts in most of the core. Modest maxima of up to 20 % sand occur around 70, 30, and 10 ka. The mean grain size of KL23 sediments is between 8 µm and 12 µm. A coarser mean is observed in the upper part of the core, at ca. 30, 23, and 7 ka. Finer mean values are found at ca. 219, 198, 128, 85, 60 and 14 ka.
XRF core scanning data of KL23 are presented in Figs. 3, S4, and S5. As for the smectite concentrations, Terr / Ca and Ti / terr show a glacial/interglacial pattern with high ratios during the glacial period, while K / terr and Al / terr show an opposite pattern.

## 4 Discussion

The northern Red Sea is surrounded by the deserts of Northeast Africa, Sinai and the Arabian Peninsula (Fig. 1) and, today, it receives most of its terrigenous sediment influx by aeolian transport. Based on air parcel backward trajectory analysis, Palchan and Torfstein (2019) suggested that the main potential source of dust supply to the northern Red Sea is northern Libya and Egypt, following the dominant NW to NNW wind system (Langodan et al., 2014). High contributions in the past from relatively inactive modern-day sources cannot be ruled out, but present-day DSAF maps (Schepanski et al., 2007; Kunkelova et al.,
2022) strongly suggest that, today, with the exception of a small coastal strip in Cyreneaica (northeasternmost Libya), the northeastern corner of Africa (Egypt and eastern Libya) is a weak source of dust in comparison to other areas in the region such as the lower latitude Eastern Sahara, the Horn of Africa, the Levant, the central Arabian Peninsula, and Mesopotamia

(see Kunkelova et al., 2024, their Fig. 9, for the most comprehensive DSAF data set). We turn, therefore, to mineralogical and geochemical techniques to determine the sources of windblown dust and riverine supply to Red Sea sediment cores.

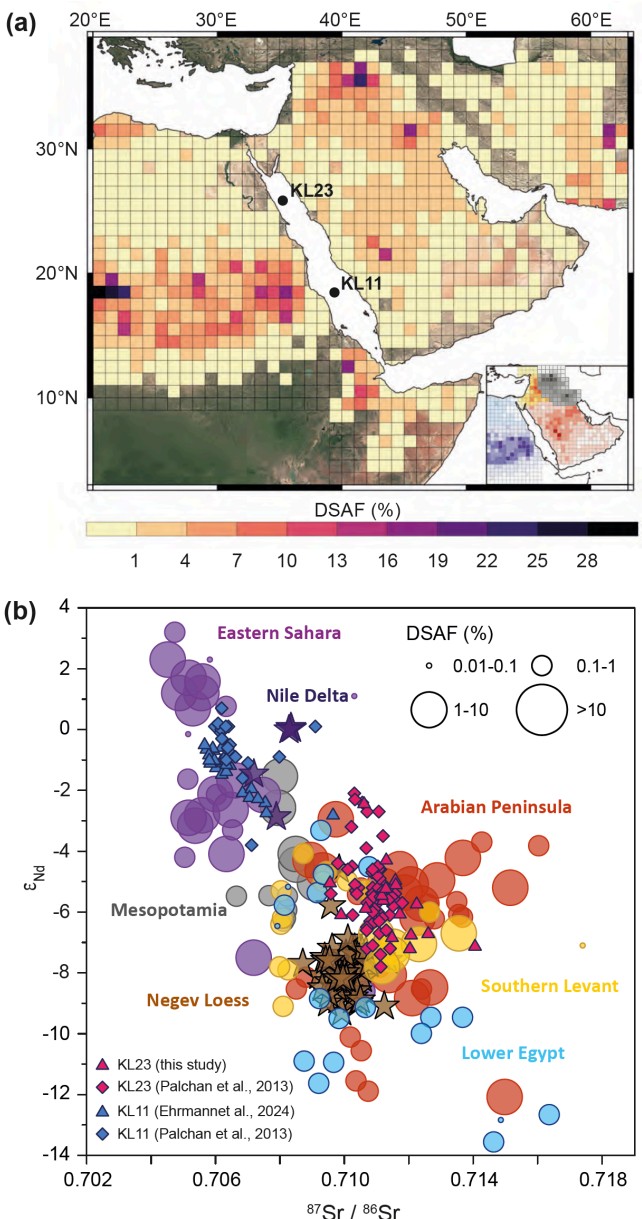

**Figure 2. Fingerprinting the provenance and transport pathway of the terrigenous fraction supplied to the northern Red Sea (core KL23) using its Sr and Nd isotope composition. (a)** Dust source activation frequency (DSAF) map of north-east Africa and south-west Asia (calculated as the percentage of days wherein one or more dust events are recorded in a 1° x 1° grid cell; after Kunkelova et al., 2022, 2024, and references therein). Background: © Google Earth. Inset shows the preferential dust source areas (PSAs) adapted from Kunkelova et al. (2022). PSAs colour coded as in panel b. **(b)** Comparison of Sr and Nd isotope data from the terrigenous fraction in marine sediment cores (triangles and diamonds) KL23 (pink) and KL11 (blue) compared to the composition of dust sources (circles). Circle size corresponds to DSAF (panel a) of sample locality. Also shown are data from shallow sediment cores on land (stars) from the Nile delta (dark purple, Fielding et al., 2018) and Negev loess (brown, Ben Israel et al., 2015).

## 4.1 Provenance of windblown dust to the Red Sea based on radiogenic isotope fingerprinting

Radiogenic isotopes provide arguably the most powerful geochemical tool for fingerprinting the provenance of terrigenous material supplied to marine sites, with the Sr- and Nd-isotope systems being the most widely used data sets. Palchan et al. (2013) reported detailed Sr and Nd isotope records for the acetic acid-insoluble residue of the <63 mm size fraction at site KL23 for late MIS 6 through early MIS 5 and late MIS 2 through MIS 1. Their Nd isotope data show excursions to relatively radiogenic values during the mid-Holocene (minimum $\varepsilon_{Nd} \sim -4.7$) and Eemian ($\varepsilon_{Nd} \sim -2.1$) AHPs, but data outside of those intervals typically fall in the range $\sim -5$ to $-8$. In intervals where there is overlap in sampling at KL23, our new Nd isotope data show excellent agreement with those of Palchan et al. (2013) (Figs. 2, 6), but the Sr isotope data of the two studies are less consistently aligned (Fig. 2b). We attribute this discrepancy in Sr isotope data to the different cleaning protocols used in the two studies. Where there is an offset, our data are more radiogenic (higher values) suggesting that KL23 sediments in these intervals host marine barite, which we removed from our samples (see Sect. 2.3), but was not removed by Palchan et al. (2013), thereby contaminating samples with Sr of a modern seawater composition ($^{87}Sr / ^{86}Sr = 0.709175$) (see Jewell et al., 2022, for details). Note that, while marine barite can severely contaminate terrigenous $^{87}Sr / ^{86}Sr$, even where barite accumulation rates are modest (Jewell et al., 2022), that is not the case at KL23. This is because, here, the offset in Sr isotope composition between the terrigenous sediments supplied to the site ($\sim$0.7095 to $\sim$0.7140) and barite (modern seawater) is modest in comparison to sites located elsewhere, for example, in the central Red Sea, the Mediterranean Sea and North Atlantic Ocean.

By comparing their Nd and Sr isotope data from KL23 to North African bedrock values, Palchan et al. (2013) inferred dust supply to the northern Red Sea from distal desert sources with more local riverine input during AHPs from the Arabian–Nubian Shield margins of the Red Sea. Our data are broadly consistent with this main interpretation. However, recent advances in the quantification of dust source activation frequency in the region and their geochemical characterization (e.g., Jewell et al., 2021, 2022; Guinoiseau et al., 2022; Kunkelova et al., 2022, 2024), allow us to move beyond relying on bedrock isotope fields to identify the distal dust sources. These new data sets were not available to Palchan et al. (2013) who inferred dust supply to the northern Red Sea from North African deserts that display central Saharan Shield granitoid values. However, our data and those of Palchan et al. (2013) are poorly aligned with the composition of any single potential African dust source active today (Fig. 2b). Instead, the radiogenic isotope composition of the terrigenous fraction in KL23 more closely matches the fingerprint of active dust sources in the central Arabian Peninsula (Fig. 2). Therefore, if we consider the radiogenic isotope data in isolation, the simplest explanation is that dust supply to the northern Red Sea basin is dominated by cold dry air outbreaks over the Arabian Peninsula that are most common in winter and drive dust-laden low level easterly jets through mountain gaps along the western margin of the peninsula (e.g., Jiang et al., 2009; Kalenderski et al., 2013; Notaro et al., 2013; Menezes et al., 2018). Alternatively, dust activation frequencies in the northeast corner of Africa were substantially higher in the past than they are today with major contributions from two or more sources mixing to yield the isotopic compositions observed in KL23. In this explanation, the main prevailing northwesterly wind systems that sweep along the basin are a major transporting agent of dust to this part of the Red Sea basin (e.g., Langodan et al., 2017; Palchan and Torfstein, 2019).

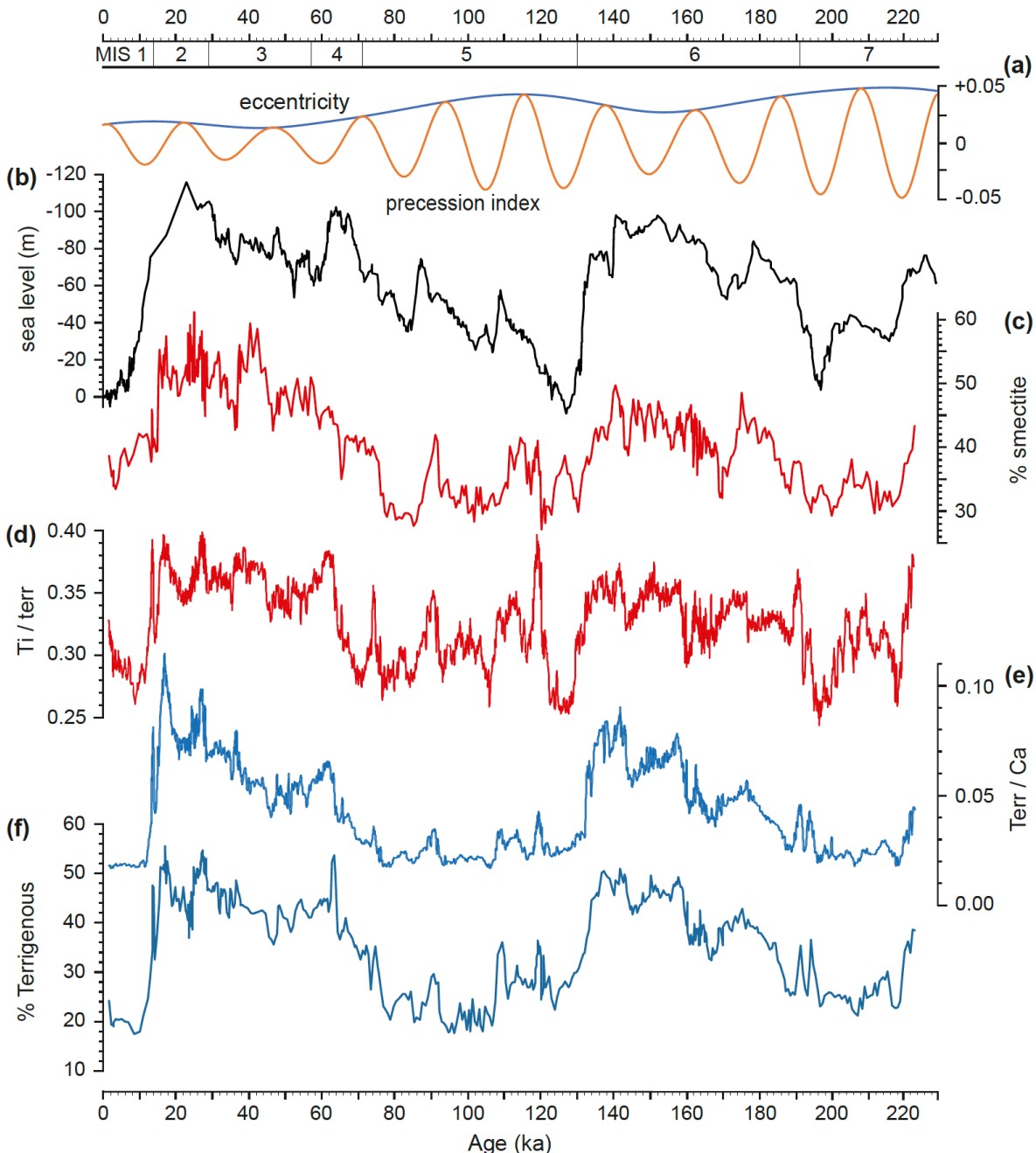

**Figure 3. Sea-level controlled aeolian sediment influx to KL23, northern Red Sea. (a) Eccentricity of the Earth´s orbit and precession index (Laskar et al., 2004). (b) Sea-level curve for the Red Sea (m, five-point running average, inverse scale; Grant et al., 2014). (c)–(f) Sediment data from KL23: (c) smectite concentration, (d) Ti / terr ratio (five-point running average), (e) Terr / Ca ratio (five-point running average), and (f) concentration of terrigenous sediment components.**

**4.2 Sea-level controlled aeolian sediment influx from the Nile delta**

The concentration of the dominant clay mineral, smectite, and the Ti / terr ratios in KL23 show similar temporal distribution patterns to each another (Fig. 3) indicating a common sediment source for smectite and Ti, rich in volcanic debris. However, only a few outcrops of Cenozoic volcanic rocks (known as harrats) exist in the direct hinterland of KL23 on the Arabian margin of the Red Sea (Fig. 1; Delaunay et al., 2024). In central Red Sea sediments (core KL11), smectite and Ti are supplied by wind from the Eastern Saharan Potential Source Area

(ESPSA) of Sudan and southernmost Egypt. This source comprises large areas containing weathering products of Cenozoic basalts, which are supplied by the Blue Nile from the Ethiopian Highlands. The smectite concentrations and Ti / terr ratios of KL11 exhibit a substantial variability in the precession band (Fig. 4). They document changes in humidity and vegetation in the ESPSA due to changing monsoon intensities (Ehrmann et al., 2024). The smectite and Ti in the northern Red Sea sediment core KL23, however, appear to be derived from a different source, because

their downcore variability shows only a minor precession-control. Instead, smectite and Ti abundances show predominantly glacial/interglacial (eccentricity) paced variability (Fig. 4) with maxima during glacials. Thus, smectite and Ti / terr indicate a main source that was highly active during glacial times of low sea level and much less active during interglacial times when sea level was high (Fig. 3).

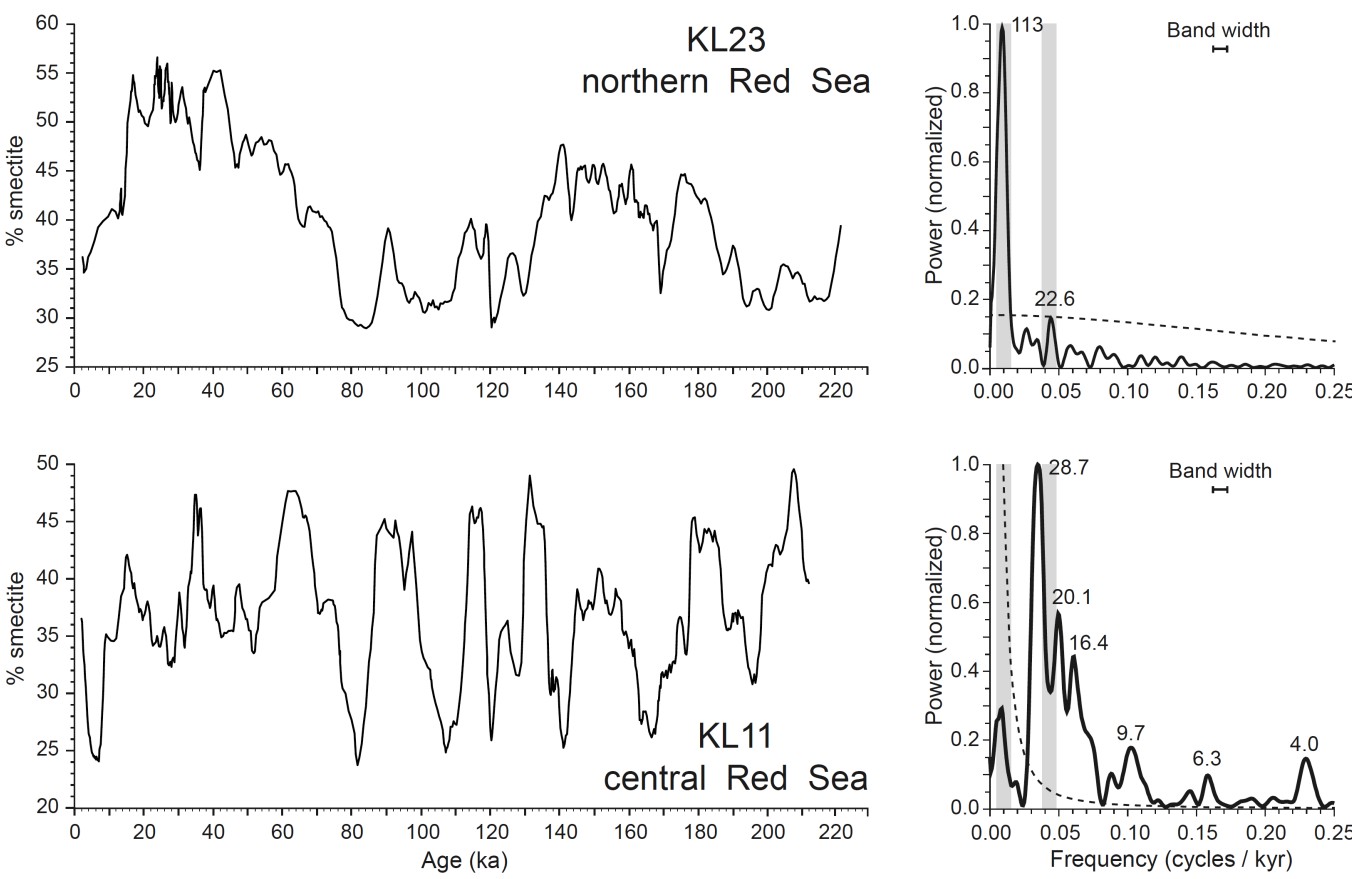

**Figure 4. Smectite concentrations (five-point running average) of cores KL23 in the northern Red Sea and KL11 in the central Red Sea, and their REDFIT spectra. Dashed lines indicate the 99 % confidence level. Significant periods in kyr are indicated. The eccentricity and precession bands are shown as grey bars.**

The fluctuations in smectite and Ti abundances are unlikely to be dominantly caused by changes in the wind system, because modelling results indicate that the direction and strength of the wind field over the Eastern Mediterranean Sea, the Levant and the northern Red Sea did not change significantly between today and the Last Glacial Maximum (Mikolajewicz, 2011; D´Agostino and Lionello, 2020).

We suggest that windblown transport from a subaerially exposed Nile delta in response to glacioeustatic sea-level fall is a viable explanation for the smectite- and Ti-rich sediments documented at KL23 during glacial conditions (Fig. 3). The delta receives abundant volcanic weathering products from the Ethiopian Highlands by the Nile (e.g., Box et al., 2011; Revel et al., 2010, 2014). Thus, delta sediments have high Ti / Al ratios (Hennekam et al., 2015) and smectite concentrations of >70 % in the clay fraction (Stanley and Wingerath, 1996). Changes in Nile sediment 255 discharge rates do not exert a strong control on aeolian sediment transport from the Nile delta to the northern Red Sea because those changes are paced by precession (e.g., Revel et al., 2010; Ehrmann et al., 2016; Bastian et al., 2021). Instead, the main control on variability in aeolian transport from the delta source is presumably deltaic subaerial exposure extent which is controlled by glacioeustatic sea-level fall. The present-day Nile delta (Fig. 1b) covers ca. 24 000 km$^2$, whereas, during the sea-level lowstands of the glacial maxima, it extended some 60–70 km 260 further seaward than today and was about 17 000 km$^2$ (ca. 75 %) larger. Nile river water discharge was lower during the glacials (Langgut, 2018; Williams, 2020) and the vegetation in the Eastern Mediterranean region was more open than today (Cheddadi and Rossignol-Strick, 1995; Langgut, 2011; Miebach et al., 2017; Duque-Villegas et al., 2024). Therefore, dry sediment could be lofted and transported to the northern Red Sea by the dominant year-round winds blowing from NW to NNW along the Red Sea axial trough (e.g., Edwards, 1987; Langodan et al., 265 2014). Similarly, parts of the SE Mediterranean shelf areas, whose sediments are governed by discharge from the Nile river (Stanley et al., 1998), also became subaerially exposed and possible dust sources. Nile delta and offshore sediments have also been considered to be likely sources of windblown supply to the Negev dunes and the Negev loess in northern Sinai (Amit et al., 2011; Muhs et al., 2013; Ben-Israel et al., 2015), the latter having a similar Nd / Sr signature to glacial-aged KL23 sediments (Negev Loess: $^{87}$Sr / $^{86}$Sr 0.7085 to 0.7114 to, $\varepsilon_{Nd}$ –11.6 to –4.6; Ben 270 Israel et al., 2015).

In keeping with the suggestion of a major contribution from the Ethiopian Highlands to Nile delta sediments, Fielding et al. (2018) report radiogenic Nd isotope values for Holocene and Pleistocene delta mudstones of between about 0 and –3 $\varepsilon_{Nd}$. Assuming the high smectite percentages of the clay fraction and the high Ti abundances in 275 KL23 sediments of glacial age indicate a distinct windblown contribution from the Nile delta to overall terrigenous

supply, if the data of Fielding et al., (2018) are representative of the subaerially exposed delta, the observed radiogenic isotope values at KL23 (typically ~ –8 to –6 $\varepsilon_{Nd}$), cannot be explained without invoking a balancing dust contribution from more unradiogenic Nd (and radiogenic Sr) source(s). While relatively dust-inactive today, lower Egypt is one candidate to solve this mass balance problem, because it lies in the path of the same prevailing NNW

winds from the Nile delta and, especially in the case of the Western Desert, shows the most unradiogenic Nd and radiogenic Sr isotopic fingerprints in the region (Fig. 2). Another possible contributor is the coastal strip of Cyrenaica which also lies upwind of the Red Sea (see the air parcel trajectory analysis of Palchan and Torfstein, 2019). There, DSAFs today are somewhat higher than the rest of north easternmost Africa (Fig. 2a). If on the other hand, the data of Bastian et al., (2021) from >1.3 km water depth on the Nile delta fan ($\varepsilon$Nd values ~ -7.5 to -2) are

representative of subaerially exposed deltaic deposits, the requirement for a balancing dust source is lessened. Furthermore, the observation that the silt fraction in fan deposits tends to carry a less radiogenic $\varepsilon$Nd signature than the clay fraction (Bastian et al., 2021), may help to explain why $\varepsilon$Nd values at KL23 (measured on the bulk terrigenous fraction) are less radiogenic than might be predicted from the high proportions of smectite in the KL23 clay fraction.

The Ti / K ratios along a N–S core transect in the northern Red Sea (Figs. 1b, S5) support the hypothesis of enhanced glacial sediment influx from a northern source dominated by volcanic debris. Maximum Ti / K ratios are reported in all cores of the transect during times of low sea level and high aeolian influx, and minimum ratios correspond to times of high sea level. Furthermore, although different types and generations of XRF core scanners were used to

scan GeoB cores and KL23, the difference between Ti / K maxima and minima follows a consistent pattern of decreasing values from N to S, with ca. 0.7 in GeoB5840, 0.6 in GeoB5833, 0.5 in GeoB5824 (Arz et al., 2001), and 0.4 in KL23 (Fig. S5). This pattern suggests that more Ti was delivered to the northern cores than to the more southern cores due to the proximity to the source.

We should also consider the possibility of a sediment delivery by ocean currents. During winter, surface waters from the central and southern Red Sea reach the region of KL23 (Cember, 1988; Yao et al., 2014), but we do not regard this as an effective delivery process for two reasons. First, the north to south decrease in Ti abundance between cores in the Red Sea (see above) and the, at times, higher smectite concentrations in core KL23 than in core KL11 in the central Red Sea imply sourcing from the north. Second, the main influx of smectite and Ti to the

central Red Sea is by aeolian transport through Tokar Gap and geological variability in this process is strongly

paced by precession (Fig. 4; Ehrmann et al., 2024), whereas our records from KL23 show glacial-interglacial timescales variability.

In addition to the Nile delta and associated shallow shelf areas, the ca. 10 000 km$^2$ large Gulf of Suez should also be considered a potential source for glacial aeolian sediments deposited in the northern Red Sea. The Gulf of Suez is <75 m deep and, therefore, dried out during some glacial intervals. Marine sediments deposited in the Gulf of Suez during interglacial periods desiccated during these glacial periods and probably were prone to wind erosion and southward transport. The only information on the clay mineral composition of sediments in the Gulf of Suez comes from its northernmost part, the Bay of Suez. Stanley et al. (1982) report mean kaolinite / smectite, illite / smectite and illite / kaolinite ratios for four sea-floor bottom samples. We converted these reported ratios to concentrations of the individual clay minerals and estimate the following: 55 % smectite, 35 % kaolinite and 10 % illite. We infer that the Gulf of Suez primarily received the smectite by aeolian transport from the Nile delta because no other source area is obvious. The higher smectite concentrations of these samples compared with near-surface samples of KL23 probably refer to the more proximal position to the source. However, we do not consider the dried-out Gulf of Suez as the likely main source for the smectite found in the northern Red Sea, because one would expect a correlation of smectite concentrations with sea level only in the range of a sea level fall of 0–75 m, the maximum depth of the gulf. The documented correlation over the full range of glacial/interglacial sea-level fall (Fig. 3), however, argues for a source in the Nile delta and adjacent shelf areas, where the area of exposed sediments changes in response to the full range of glacial/interglacial sea-level change.

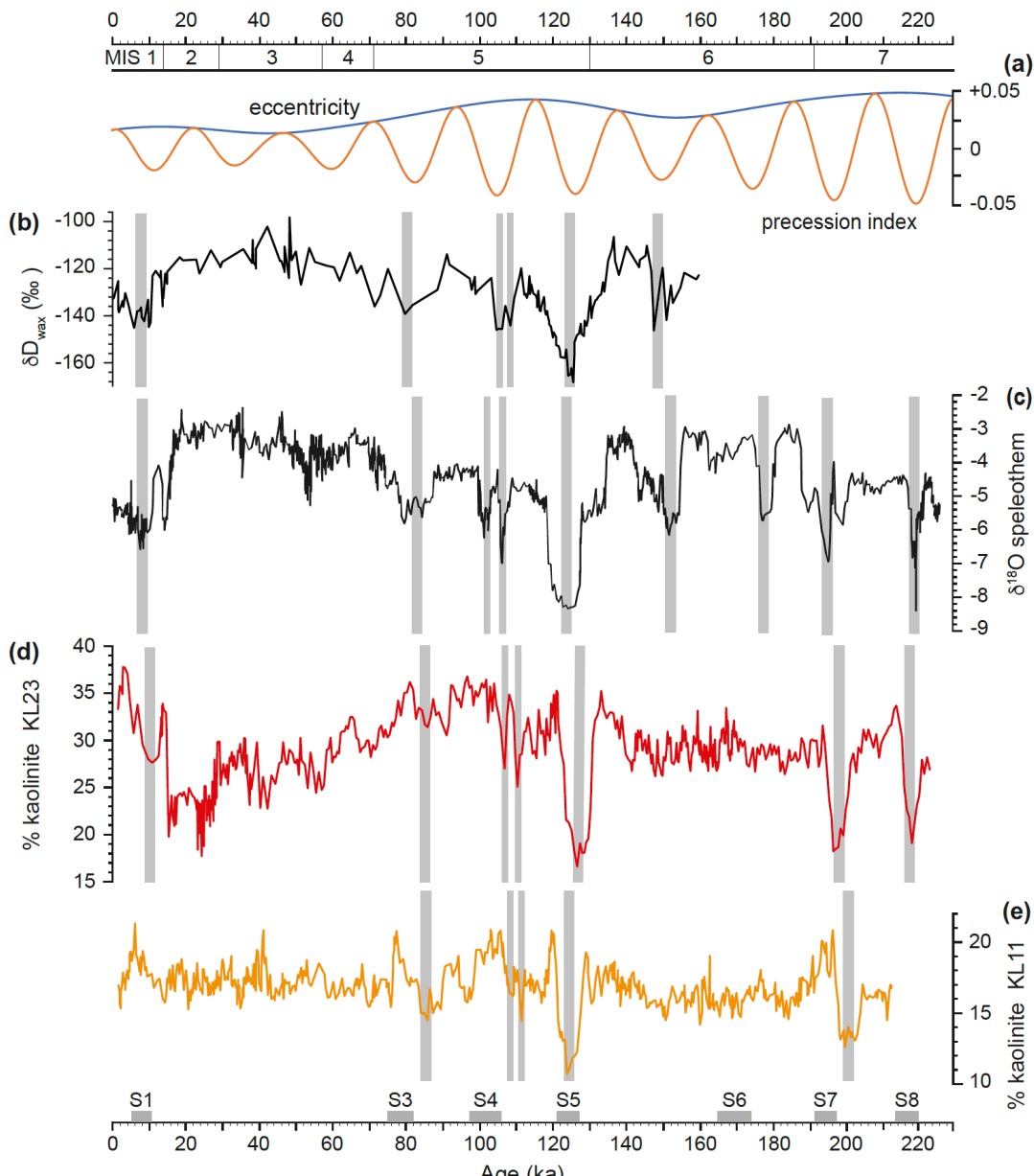

**Figure 5. Record of dry and humid phases in the Levant and hydrologically controlled aeolian sediment influx to the Red Sea. (a) Eccentricity of the Earth´s orbit and precession index (Laskar et al., 2004). (b) Leaf wax δD isotope record (‰ VSMOW) in sediments drilled in the Dead Sea (Tierney et al., 2022). (c) Combined speleothem δ$^{18}$O record from Soreq Cave (0–184 ka) and Pequin Cave (>184 ka) (Bar-Matthews et al., 2003). Concentrations of kaolinite (d) in core KL23 from the northern Red Sea and (e) in core KL11 from the central Red Sea. Kaolinite minima indicate humid periods. The main humid periods are shown by vertical grey bars. Marine isotope stages (MIS) are shown at the top; horizontal bars at the bottom indicate sapropel layers S1–S8 in the eastern Mediterranean Sea associated with the African Humid Periods.**

## 4.3 Hydrologically controlled aeolian sediment influx

The downcore variability of the other clay minerals in core KL23 differs from that of smectite (Fig. S2), demanding different sources and different control mechanisms. Kaolinite abundance shows less overall downcore variability than smectite, but marked short-term reductions at ca. 218, 198, and 128 ka, i.e. at times of AHPs 8, 7, and 5. Sediment core KL11 in the central Red Sea shows a similar kaolinite pattern, but with lower concentration levels than for KL23 (Fig. 5; Ehrmann et al., 2024). We infer a main kaolinite provenance from Egypt and Sinai, where high concentrations of kaolinite have been reported in weathering products originating from mainly Cretaceous and Cenozoic sedimentary rocks (compilation by Hamann et al., 2009). In this interpretation, the kaolinite is transported along the Red Sea axis by the same general northerly wind system as the smectite from the Nile delta and the kaolinite minima recorded during AHPs 8, 7, and 5 indicate vegetation cover in the source region in response to enhanced humidity caused by a southward shift of the Mediterranean winter precipitation system (Kutzbach et al., 2020; Cheddadi et al., 2021). While the main source of smectite influx to KL23 is inferred to have been sourced from the Nile delta, our records also show (subordinate) concentration maxima coinciding with AHPs (Fig. S6). Enhanced humidity is also documented in other proxy data from the region, such as the isotopic composition of plant waxes in sediment cores from the eastern Mediterranean Sea (Meyer et al., 2024) and the Dead Sea (Tierney et al., 2022; Fig. 5), and the speleothem oxygen isotope record from Soreq Cave and Pequin Cave (Bar-Matthews et al., 2003; Fig. 5). In general, speleothem records indicate a strong N–S precipitation gradient. Whereas in northern and central Israel, speleothems grew continuously and indicate sufficient humidity during both glacial and interglacial periods, records in the central and southern Negev as well as northern Africa indicate only short humid periods with speleothem growth during peak interglacials (Vaks et al., 2013; Bar-Matthews, 2019; El-Shenawy et al., 2018). Thus, the Mediterranean rainfall system and the African monsoon are closely coupled, with an intensification and southward extension of the Mediterranean rainfalls in winter coinciding with strong African summer monsoon activity (c.f., Kutzbach et al., 2014; Wagner et al., 2019; Blanchet et al., 2021; Tierney et al., 2022; Meyer et al., 2024).

The illite and palygorskite concentrations in KL23 sediments correlate roughly with each other (Fig. S2), and the two clay minerals are assumed to come from a common source. We hypothesise that their main source is situated on the eastern Arabian Peninsula and/or Mesopotamia, as discussed in detail by Ehrmann et al. (2024). Presumably, this source is also expressed by high K / terr and Al / terr ratios (Fig. S4), which exhibit a distribution pattern opposite to Ti / terr and indicate a source dominated by basement rocks such as those of the Arabian–Nubian Shield. Mesopotamia and the eastern Arabian Peninsula are well known as very active dust emission centres at present time (e.g., Ramaswamy et al., 2017; Kunkelova et al., 2022). The Arabian Peninsula was relatively wet during earlier interglacial periods (e.g., Nicholson et al., 2020; Dallmeyer et al., 2020, 2021; Groucutt et al., 2021), but rainfall is inferred to have been mainly concentrated in the summer months. The northern and northeastern Arabian Peninsula maintained desert conditions (Jennings et al., 2015; Dallmeyer et al., 2020, 2021) allowing continued dust uptake. The concentrations of illite and palygorskite in core KL23 are substantially lower than those of smectite, and their temporal distribution patterns are likely influenced by dilution with smectite. In the central Red Sea core KL11, the

illite and palygorskite concentration patterns are also affected by dilution with smectite, there, however, on a precession time scale, because smectite influx was controlled by the monsoon intensity (see above; Ehrmann et al., 2024). This confirms a persistent dust activity in Mesopotamia and/or on the northern Arabian Peninsula. The dust may be transported westward by the low-level jets that blow today mainly during winter through mountain gaps towards the northern Red Sea (Jiang et al., 2009; Menezes et al., 2018). In summary, while smectite concentration is mainly controlled by the sea level-led availability of

the inferred source area, the desiccated Nile delta, and therefore reflects glacial-interglacial cycles, the abundances of other wind-transported clay minerals are controlled by humidity and vegetation cover in their source areas.

## 4.4 Fluvial sediment influx and its control

The hydroclimate of North Africa has oscillated between arid and humid conditions on astronomical timescales for at least 11

million years (Crocker et al., 2022). The AHPs were driven by an intensification and northward expansion of the monsoon rains during summer, due to solar insolation changes and feedback mechanisms (e.g., Rossignol-Strick, 1983; Gasse, 2000; Tjallingii et al., 2008; Enzel et al., 2015; Grant et al., 2017; Drake et al., 2022; Armstrong et al., 2023). Local expressions of the AHPs are also recorded by fluvial sediments of the central Red Sea (Palchan et al., 2013; Ehrmann et al., 2024). Even north of the main palaeo-monsoon activity, in the northern Red Sea, humid periods with local floods are documented (Arz et al.,

2003; Palchan et al., 2013; Palchan and Torfstein, 2019; Hartman et al., 2020).

Present-day rainfall of >100 mm/a is confined to a narrow strip along the Mediterranean coast. It decreases to <50 mm/a in the northernmost Red Sea. Rainfall in the Eastern Mediterranean and the Levant is associated with eastward migrating Mediterranean winter cyclones (Eshel, 2002; Ziv et al., 2006; Enzel et al., 2015). However, the Mediterranean cyclones bring

little rain to Arabia (Enzel et al., 2015). Precipitation is also provided by tropical plumes approaching during winter and spring. The plumes originate in tropical Africa and move northeastward across northern Africa towards the Eastern Mediterranean Sea and the northern Arabian Peninsula. They may bring rainfall and flooding to eastern Egypt, Sinai, Negev and northern Arabia (Rubin et al., 2007; Enzel et al., 2015). Furthermore, the semi-permanent low-pressure trough over the Red Sea (Red Sea Trough) can create intense local rainfall and floods, mainly during autumn, over the southern Levant and the western and

northern Arabian Peninsula (de Vries et al., 2013; Enzel et al., 2015).

No major perennial rivers currently drain into the northern Red Sea. However, short wadis, mainly incised into the western margin of the Arabian Peninsula, testify to former fluvial episodes. Wadi al Hambd is by far the largest of these drainage systems (Fig. 1b). Its NW-SE stretching catchment area exceeds 100,000 km$^2$, has a length of ca. 700 km and a width of up to

200 km. Its mouth lies at ca. 25.5°N, just east of KL23. The catchment is mainly composed of Precambrian rocks of the Arabian–Nubian Shield, consisting dominantly of metasedimentary and metavolcanic rocks, and some Cenozoic formations including basalts (El Maghraby et al., 2014; Delaunay et al., 2024).

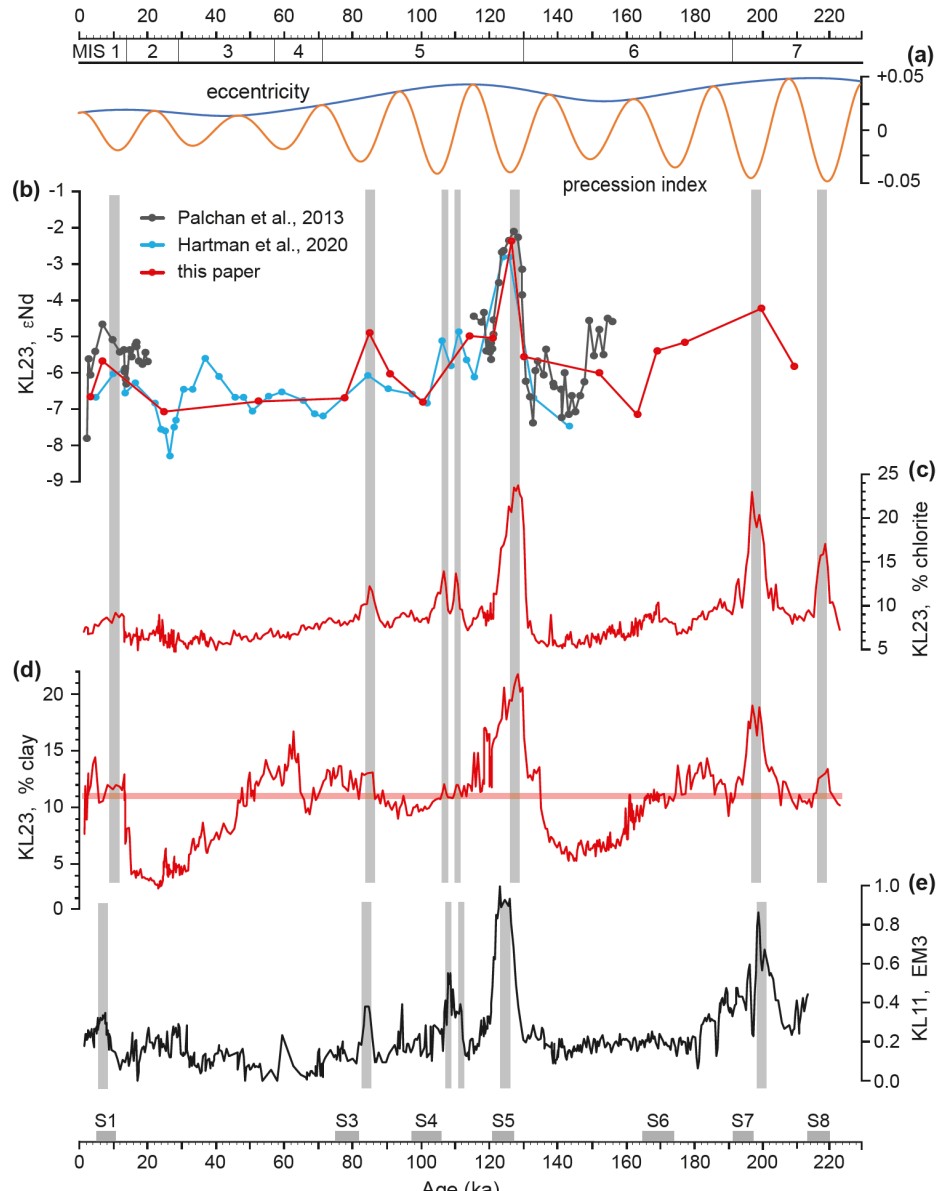

**Figure 6.** Fluvial sediment influx to the Red Sea during the last ca. 220 kyr as indicated by (a) Eccentricity of the Earth´s orbit and precession index (Laskar et al., 2004). (b) $\varepsilon_{Nd}$ data from sediments of KL23 (data from this study; Palchan et al., 2013; Hartman et al., 2020), (c) chlorite concentration, KL23 (this study), (d) concentration of the terrigenous clay fraction (<2 µm), KL23 (this study), and (e) loadings of the fluvial grain size endmember EM3 in core KL11, central Red Sea (Ehrmann et al., 2024). The main humid periods as documented by KL23 and KL11 are shown by vertical grey bars. Marine isotope stages (MIS) are shown at the top; horizontal bars at the bottom indicate sapropel layers S1–S8 in the eastern Mediterranean Sea associated with the African Humid Periods.

Fluvial episodes are documented in Red Sea sediments by increased clay content in the terrigenous sediment fraction, enhanced terrigenous sedimentation rates, changes in sediment composition and a lower sea surface salinity (Arz et al., 2003; Palchan et al., 2013; Palchan and Torfstein, 2019; Hartman et al., 2020; Ehrmann et al., 2024). Geologically short-term episodes of fluvial sediment influx to KL23 are recorded by enhanced concentrations of clay-sized terrigenous sediment components, although part of the clay is contained in the aeolian dust, and by chlorite and $\varepsilon_{Nd}$ maxima (Fig. 6). The general level of terrigenous clay abundance in KL23 sediments is about 10 %. Lower concentrations are restricted to the last two glacial maxima, when climate was especially dry and fluvial influx was likely at a minimum. Higher concentrations of up to 20 % probably indicate fluvial discharge from the Red Sea borderlands. According to the grain size data, the most humid period occurred at ca. 128 ka, others at 218, 198, and 10 ka (Fig. 6). Broader but less intense clay maxima occur around 80 ka and 60 ka.

More detailed information comes from the chlorite record, which documents well-defined local flood events (Fig. 6). Chlorite is most likely derived from the weathering of metavolcanic rocks, greenschists and chlorite schists cropping out within the Arabian–Nubian Shield in the uplifted Red Sea borderlands (USGS, 1963; EGSMA, 1981; Brown et al., 1989; El Kalioubi et al., 2020). Sharp chlorite maxima occur, in decreasing order, at 128, 198, 218, 106, 111 ka, 85, and 10 ka. In consideration of uncertainties in the different age models, these fluvial events occur at the same time and in the same intensity as the AHPs 5, 7, 8, 4a, 4b, 3 and 1, respectively, as documented by sapropels in the Eastern Mediterranean Sea (e.g., Emeis et al., 2003; Ehrmann and Schmiedl, 2021). A very similar record of fluvial activity is documented in the monsoon-controlled sedimentary sequence of core KL11 from the central Red Sea (Fig. 6; Ehrmann et al., 2024). The humid episodes at 111, 106, and 85 ka are not recorded in the clay content data at KL23 (Fig. 6), probably because the clay content is not only a proxy for river discharge, but instead represents a combination of aeolian and fluvial transport. As in other sedimentary records, both from the Eastern Mediterranean Sea and the central Red Sea (Ehrmann and Schmiedl, 2021; Ehrmann et al., 2024), no fluvial activity is reported in KL23 for the glacial AHP6 at ca. 170 ka, probably because the glacial climate conditions suppressed the influence of the monsoon and other rain-bringing climate systems in the Red Sea region.

In the middle of the humid phases corresponding to AHP5 and AHP7 subordinate smectite maxima occur both in core KL23, northern Red Sea, and KL11, central Red Sea (Fig. S6). Although our smectite record for KL23 is generally interpreted to represent aeolian dust supply from the Nile delta, the subordinate maxima for AHP5 and AHP7 correlate with maxima in proxies for fluvial influx, such as chlorite concentrations and fine grain sizes (Figs. 3, 6). Thus, we infer their origin through intense chemical weathering of volcanic rocks in the headwaters of wadis, and fluvial transport to the Red Sea during peak humid intervals. This interpretation may also help to explain the peaks in $\varepsilon_{Nd}$ during these AHPs although we note that the metasedimentary and metavolcanic Precambrian rocks of the Arabian–Nubian Shield are themselves considered sufficiently radiogenic to explain the riverine-derived $\varepsilon_{Nd}$ peak documented for AHP5 (Palchan et al., 2013). We also note that the smectite signal is less pronounced in AHP7 than in AHP5, implying that AHP7 was less intense. No corresponding smectite signals

occur in the other AHPs, suggesting they were not sufficiently intense to drive smectite generation and transportation in this way and/or the signal was swamped by windblown smectite derived from the subaerial exposure of the Nile delta. Palaeo-drainage systems are incised into both the African and Arabian margins of the northern Red Sea where they cut through the basement rocks of the Arabian–Nubian Shield (e.g., USGS, 1963; EGSMA, 1981). Our chlorite and smectite records for AHP5 and AHP7 from KL23 are very similar to those from KL11 in the central Red Sea (Fig. S6). There, the source of fluvial input during AHPs is clearly the African margin, down the Baraka River system, which today is only ephemerally active (Ehrmann et al., 2024). On this basis, fluvial input from the African margin to KL23 during AHP5 and 7 may be implied. However, by far the most prominent of the ancient riverine systems feeding into the northern Red Sea is Wadi Al Hambd which lies on the Arabian margin and, unlike its African counterparts, its headwaters drain from shield-capping young volcanic rocks (Fig. 1) or 'harrats'. These volcanics represent the most obvious local source of smectite and an even more radiogenic source of Nd than the rocks of the Arabian–Nubian Shield. This observation may explain why the response of $\varepsilon_{Nd}$ in the terrigenous fraction to the transition from marine isotope stage 6 (glacial conditions) to AHP5 is more pronounced at KL23 than KL11 (Fig. 6) (Ehrmann et al. 2024; Palchan et al., 2013) and points towards an Arabian provenance for fluvial input to KL23 during these AHPs.

Regardless, despite the different climate systems, we see the same relative intensities of northern Red Sea humid phases as in the African summer monsoon rains documented in the Eastern Mediterranean Sea and the fluvial influx episodes to KL11 in the central Red Sea (Ehrmann and Schmiedl, 2021; Ehrmann et al., 2024; Fig. 6). Our results do not distinguish between the different potential mechanisms responsible: intensification of Mediterranean cyclones, the Red Sea Trough and/or tropical plumes are all viable. Nevertheless, our findings do point to coupled change in hydroclimate on geological timescales between our study region where rainfall today is dominated by winter rainfall and the region to the south where rainfall is controlled by the summer monsoon.

## 5 Conclusions

We investigated the composition of deep-sea sediments retrieved in core KL23 from the northern Red Sea in order to reconstruct sediment provenance and to decipher controlling mechanisms of aeolian and fluvial sediment influx throughout the last ca. 220 kyr. Based on our study the following conclusions are drawn:

1.  Terrigenous sediments deposited of the northern Red Sea are sensitive recorders of changes in the intensity of late Quaternary aeolian and fluvial sediment influx.

2.  Our smectite and Ti records indicate a windblown dust source in the Nile delta characterized by volcanic weathering products derived from the Ethiopian Highlands. The uptake of this dust is strongly controlled by sea level. At times of

high sea level, as today, this source was not active. At times of low sea level during the glacial periods, dust uptake was at a maximum due to subaerial exposure of a large Nile delta, more open vegetation and arid climate.

3.      Kaolinite characterizes dust originating mainly from Egypt and Sinai. Downcore changes in the kaolinite concentration are controlled by hydroclimate variability in this region. Geologically short-term minima in kaolinite concentration during AHPs 8, 7, and 5 indicate a reduced dust export during times of intensified winter rainfall, presumably due to an enhanced Mediterranean cyclogenesis and a southward expansion of the Mediterranean rain belt.

4.      Geologically short-term maxima in chlorite concentrations, fine grain sizes of the terrigenous sediment components and high $\varepsilon_{Nd}$ document episodes of fluvial influx to the northern Red Sea. They correlate with decreased aeolian dust influx from the north. The most likely candidate for fluvial supply is the Wadi al Hambd on the Arabian Peninsula.

5.      The humid phases documented by the reduced influx of aeolian dust and the enhanced influx of fluvial sediments correlate with the well-known AHPs, not only in timing but also in relative intensity. In decreasing order of intensity, these episodes correspond to AHP5 (ca. 128 ka), AHP7 (ca. 198 ka), AHP8 (ca. 218 ka), AHP4 (ca. 111 ka and 106 ka), AHP3 (ca. 86 ka), and AHP1 (ca. 10 ka). Thus, there seems to be a strong link between the climate systems of the African monsoon, the Mediterranean cyclones, and/or possibly tropical plumes and the Red Sea Trough.

**Data availability**

All data will be made accessible via the PANGAEA database at the Alfred Wegener Institute for Polar and Marine Research, Bremerhaven, Germany (https://doi.pangaea.de/10.1594/PANGAEA.975131, Ehrmann et al., 2025).

**Author contributions**

WE and GS initiated and designed the research project. WE was in charge of the sedimentological data. PAW was in charge of the radiogenic isotope data. All authors contributed to the interpretation and discussion of the data and participated in writing the submitted and the revised manuscript.

**Competing interests**

The authors declare that they have no conflict of interest.

**Acknowledgements**

We thank the master and the crew of "RV *Meteor*", the chief scientist Peter Stoffers (Kiel) and the group leader Christoph Hemleben (Tübingen) for their efforts during cruise M31/2 in 1995. We also thank the core curator Hartmut Schulz (Tübingen). Werner Ehrmann and Gerhard Schmiedl are grateful to Sylvia Haeßner for her excellent performance in the sedimentological laboratories at the University of Leipzig. Dennis Bunke did the grain size analyses and the XRF measurements. Furthermore, we acknowledge the manifold scientific and technical support from Stefan Krüger. The study is a contribution to the Cluster of Excellence "CLICCS - Climate, Climatic Change, and Society", and a contribution to the Center for Earth System Research and Sustainability (CEN) of the University of Hamburg. Paul A. Wilson thanks Kai Zhang, Anya Crocker and Tereza Kunkelova for helpful discussions and Amelia Gale, Yuxi Jin, Matt Cooper and Andy Milton for laboratory assistance. We are grateful to Daniel Palchan and Carlo Mologni for their reviews. They provided useful comments and suggestions for improving the manuscript.

**Financial support**

The German Research Foundation (Deutsche Forschungsgemeinschaft, DFG) financially supported the studies of Werner Ehrmann and Gerhard Schmiedl (grant nos. Eh 89/23-1, Schm 1180/26-1). Paul A. Wilson was supported by the Royal Society (Challenge Grant CH160073 and Wolfson Merit Award WM140011), NERC (grant no. NE/X000869/1), and University of Southampton's GCRF strategic development grant 519016. The "Open Access Publishing Fund" of Leipzig University supported by the German Research Foundation within the programme "Open Access Publication Funding" covered the publication costs.

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
