# Peer review of "Controls of aeolian and fluvial sediment influx to the northern Red Sea over the last 220 000 years"

_Climate of the Past, 2024_

## Referee Comment (RC1)

Review for: Controls of aeolian and fluvial sediment influx to the northern Red Sea over the last 220 000 years

Werner Ehrmann1, Paul A. Wilson2, Helge W. Arz3, Gerhard Schmiedl4

The manuscript is original and provides detailed information regarding the siliciclastic sediment compositions in core KL23 from the north part of the Red Sea. The data set provided in this manuscript is valuable and continues this group's work from recent years, where they infer paleoclimate trends from the isotopic values and clay minerals compositions. This contribution is important in providing high-resolution mineralogical and geochemical data and should be published for others to use. The discussion uses the available literature and draws broad spatial teleconnections based on various paleoclimate records and models. They provide a strong case for the ties between low latitude northern Red Sea and high latitudes ice caps glacial-interglacial cycle climate variability over the equatorial insolation driven variability seen southward in Red Sea archives.

The authors argue for a reasonable scenario – where during glacial periods and low global sea levels, the Nile River delta was exposed, and its sediments served as a significant source for terrigenous eolian sediments blown southward to the Red Sea. Their argument relies on increased smectite content and Ti counts during glacial periods, both likely originating from volcanic detritus. Another source of eolian sediments suggested by the authors to be significant in the past is the "Tokar Gap" and two other similar mountain gaps in the eastern borders of the Red Sea fringing mountain belt. These interpretations of the results might prove valid, however, other interpretations may well be inferred from the same results, following the discussion raised here:

**Line 155 –** contrary to the stated argument the DSAF% maps from (Kunkelova et al., 2024) shows relatively high values for the region between Sallum (Egypt) and Benghazi (Libya). Indeed, this region is the source of reconstructed air parcel routes (Palchan and Torfstein, 2019). On the other hand, lower latitude East Sahara is not a probable source of dust to KL23 due to the local wind patterns and their convergence southward from it (e.g., Menezes et al., 2018).

**Line 180 –** the treatment of removing marine barite is important but seems not very significant in interpretation of the provided Sr isotopes, as all of the previous terrigenous data from KL23 (Palchan et al., 2013) is higher than the modern seawater composition of 0.709. Hence, as to the authors claim, it should be even more radiogenic than reported. Even so, comparing the Sr values in the current and previous work the difference seems to be negligible.

**Line 258 –** using the term "substantial" is a bit of a stretch as core KL23 smectite base levels are around 40% of the clay composition, thus, the rise during glacial periods is only additional 10%. This increase is proportional to the content of other clays as the analysis was done only on the <2um fraction. Thus, the rise could reflect decrease in other clays rather than more input from a specific source.

Furthermore, the concentration of the clay fraction in the samples drops significantly from ±12% to ±4% during the respective interval of increased smectite (Fig. 3B & Fig. 6C). However, the use of εNd reflects sources without this issue and its low values "(typically ~ –8 to –6 εNd)" points to that if there is indeed a Deltaic source, it is surely not "substantial" as it resembles more granitoid detritus compared with the Deltaic higher εNd values.

Methods remarks:

Section 2.3 – the leaching method is not specified. This is important and needs clarification and detail. Similarly, there is no detail on the analysis method (i.e., TIMS? Multi-collector?). Even if this is described in a previous paper, it is important to include minimal information regarding the method and analysis (indeed, this is discussed later in section 4.1). For example, what standard was used during the analysis, and what value was assumed for it?

Figure remarks:

**Fig. 2a** the window lacks a crucial potential source area depicted as increased DSAF% in northern Sahara around 20°N (Kunkelova et al., 2024). This region is a prominent source of air parcel reconstruction (Palchan and Torfstein, 2019).

In summary, this is a fascinating paper with substantial data contribution on the clay mineralogical compositions in the northern Red Sea – a region largely overlooked. The conclusions drawn based on the results are partly debatable; the paleoclimate community will surely benefit from the discussion.

Daniel Palchan

**References**

Kunkelova T., Crocker A. J., Wilson P. A. and Schepanski K. (2024) Dust Source Activation Frequency in the Horn of Africa. *J. Geophys. Res. Atmospheres* **129**, e2023JD039694.

Menezes V. V., Farrar J. T. and Bower A. S. (2018) Westward mountain-gap wind jets of the northern Red Sea as seen by QuikSCAT. *Remote Sens. Environ.* **209**, 677–699.

Palchan D., Stein M., Almogi-Labin A., Erel Y. and Goldstein S. L. (2013) Dust transport and synoptic conditions over the Sahara–Arabia deserts during the MIS6/5 and 2/1 transitions from grain-size, chemical and isotopic properties of Red Sea cores. *Earth Planet. Sci. Lett.* **382**, 125–139.

Palchan D. and Torfstein A. (2019) A drop in Sahara dust fluxes records the northern limits of the African Humid Period. *Nat. Commun.* **10**.

---

## Referee Comment (RC2)

The manuscript presents original research on siliciclastic sediment compositions in core KL23 from the northern Red Sea. The dataset is valuable as it extends the authors' previous work on paleoclimate trends through isotopic values and clay minerals. This study provides exceptionally high-resolution mineralogical and geochemical data supporting hypothesis on wind transport circulation between the Lower-Nile valley and the northern Read Sea over ~220 ka.

The discussion effectively integrates literature and establishes connections between northern Red Sea climate variability and glacial-interglacial cycles in high-latitude ice caps, contrasting with equatorial insolation-driven changes further south. The authors argue that during glacial periods and low sea levels, the exposed Nile River delta was a key source of eolian sediments, as indicated by increased smectite content, Ti counts and high εNd values.

However, some discrepancies exist between the data and the presented hypothesis. These discrepancies are not adequately explained, nor do the authors open the discussion to alternative hypotheses that deserve consideration. The following sections—GENERAL QUESTIONS, GENERAL COMMENTS, DETAILED COMMENTS, and FIGURE COMMENTS—highlight these issues.

**GENERAL QUESTIONS**

1) If, as hypothesized by the authors, the smectite fraction originates from the radiogenic Nile Delta sediments (average εNd ≈ -3) exposed during low sea level periods, why do high smectite and Ti concentrations during the Last Glacial Maximum (LGM) correspond to extremely low εNd (~ -8), which are characteristic of non-Nilotic sources, closer to the Saharan Shield?

2) If smectite is associated with radiogenic Nile Delta sediments, as the authors suggest, why do low smectite values during the S5 period correspond to high εNd values (-1)? The authors interpret this period as one dominated by increased local sediment supply (chlorite). Does this imply that the northern Red Sea is also influenced by highly radiogenic local (non-aeolian) sources?
It is worth noting that the eastern margin of the northern Red Sea consists of recent (Oligocene to Quaternary) volcanic headwaters, which can serve as sources of smectite and high εNd radiogenic values (see:

*Antoine Delaunay, Guillaume Baby, Evelyn Garcia Paredes, Jakub Fedorik, Abdulkader M. Afifi, Evolution of the Eastern Red Sea Rifted Margin: Morphology, Uplift Processes, and Source-to-Sink Dynamics, Earth-Science Reviews, Volume 250, 2024, 104698, ISSN 0012-8252, https://doi.org/10.1016/j.earscirev.2024.104698).*

3) If major and perennial fluvial sediment supply to KL23 is excluded, as proposed by the authors, the observed sedimentation rates appear disproportionately high compared to accumulation rates in the Nile Delta. This is particularly striking if KL23 sediments are assumed to be exclusively of aeolian origin.

How do the authors explain that aeolian sedimentation rates in the Red Sea are equal to or even higher than many fluvial sedimentation rates?

These discrepancies remain unresolved. To address these issues, I encourage the authors to expand the discussion by considering additional hypotheses based on the data (see GENERAL COMMENTS below).

**GENERAL COMMENTS**

**A) The Gulf of Suez as a Sediment Source**

Based on source proxies (smectite and εNd ), the authors suggest that most of the KL23 sediment originates from Aeolian-reworked dust from the exposed Nile Delta during low sea level periods. However, none of the presented data directly confirm an eolian origin (e.g., grain surface analysis via exoscopy or grain-size distribution analysis).

The Gulf of Suez serves as a sediment repository for particles transported by marine currents from the Nile River. Therefore, high-smectite, radiogenic εNd sediments could simply originate from the erosion of the Gulf of Suez continental shelf during low sea level periods. The hypothesis that the Gulf of Suez serves as a temporary, non-linear reservoir for high-smectite and radiogenic εNd sediments could provide a plausible explanation for Questions 1 and 2.

**B) The Role of Shallow and Deep-Water Circulation in the Red Sea**

The manuscript by Ehrmann et al. thoroughly discusses wind circulation around the study area, treating it as the main transport mechanism for clay particles at the KL23 site. However, it does not consider shallow or deep-water Red Sea circulation as a potential transport mode for smectites and radiogenic εNd sediments from the central/southern Red Sea.

As shown by **Yao et al. (2014)**, shallow waters originating from the central/southern Red Sea reach the KL23 site (~25°N). These waters carry hydro-sedimentary inputs from the Eritrean/Ethiopian Basaltic Traps headwaters. Around 24°–25°N, sinking processes induce downwelling, potentially transporting sediment plumes rich in smectites and radiogenic εNd particles from the Barka River and other sources in Eritrea.

Please consider and develop this hypothesis in the discussion.

[Figure]

**Figure 14.** Schematic for the three-dimensional overturning circulation in the northern Red Sea. Most (0.5 Sv) of the surface western boundary current (0.6 Sv) crosses the basin at around 24°N, and then either sinks along the eastern boundary at the crossing latitude (0.1 Sv) or switches to an eastern boundary current (0.4 Sv) and sinks along the eastern boundary through a cyclonic recirculation. The downwelled water at the intermediate depth is transported to the western boundary either through direct cross-basin flows or a rim current along the boundary. Meanwhile, the sinking along the eastern boundary is enhanced by a weaker cross-basin overturning circulation produced by the upwelling along the western boundary (0.2 Sv). A small portion of the western boundary current (0.1 Sv) sinks in the Gulf of Aqaba and Gulf of Suez and contributes to the intermediate and deep water.

Reference:

Yao, F., Hoteit, I., Pratt, L. J., Bower, A. S., Kohl, A., Gopalakrishnan, G., & Rivas, D. (2014). Seasonal Overturning Circulation in the Red Sea: 2. Winter Circulation. J. Geophys. Res. Oceans, 119, 2263–2289. doi:10.1002/2013JC009331.

**C) Sedimentation Rates and εNd Variability between the Nile Delta and KL23**

Sedimentation rates at KL23 are notably high compared to those in the Nile Delta. Similarly, the average εNd values often overlap with those from Nile Delta coring sites. Additional data supporting and discussing source correlations with the study site would be beneficial. For example, a 100-ka-long dataset of εNd , smectite, and sedimentation rates from the Nile Deep Delta Fan is available in:

*Luc Bastian & Carlo Mologni, Nathalie Vigier, Germain Bayon, Henry Lamb, Delphine Bosch, Marie-Emmanuelle Kerros, Christophe Colin, Marie Revel, Co-variations of Climate and Silicate Weathering in the Nile Basin during the Late Pleistocene, Quaternary Science Reviews, Volume 264, 2021, 107012, ISSN 0277-3791,* [https://doi.org/10.1016/j.quascirev.2021.107012](https://doi.org/10.1016/j.quascirev.2021.107012).
* * *
**DETAILED COMMENTS**

**Line 26:** Specify which grain size fraction is being analyzed.

**Line 80:** Provide a synthetic figure of the age model as a review of all ages given in previous works. The manuscript also needs a presentation of sedimentation rates in the sequence.

**Line 91:** Include a detailed protocol in the supplementary materials.

**Line 102:** Explain why these elements are representative of the terrigenous sediment fraction, with mineralogical justification or bibliographic support.

**Line 141:** "Modest maxima of up to 20%"—20% of what? Please specify.

**Line 215:** There are multiple volcanic sources on the Arabian headwaters of the Red Sea; consider citing:

*Antoine Delaunay, Guillaume Baby, Evelyn Garcia Paredes, Jakub Fedorik, Abdulkader M. Afifi, Evolution of the Eastern Red Sea Rifted margin: morphology, uplift processes and source-to-sink dynamics, Earth-Science Reviews, Volume 250, 2024, 104698, ISSN 0012-8252,* [https://doi.org/10.1016/j.earscirev.2024.104698](https://doi.org/10.1016/j.earscirev.2024.104698).

**Line 216:** The Barka River, originating in the Eritrean Highlands over a basaltic trap context, flows northward into the Red Sea (~640 km in length). Given the northward shallow water circulation in the Red Sea (see GENERAL COMMENT B), its potential recent volcanic sediment contributions to the northern Red Sea should be considered.

**Line 379:** Specify what "fine grain size" refers to.
* * *
**FIGURE COMMENTS**

**Figure 1a:** Please add εNd  values to the map, corresponding to geological regions in Africa and the Arabian Peninsula.

**Figures 4 & 5:** Add precession and eccentricity curves to support hypotheses discussed in the main text. Correlating εNd, smectite, and sedimentation rates with Nile Delta sequences is strongly recommended.

This review would  enhance the text's clarity and introduces additional plausible hypotheses that should be considered in this article.

Carlo Mologni

---

## Author Comment (AC1)

**Review by Daniel Palchan**

Responses in red

Review for: Controls of aeolian and fluvial sediment influx to the northern Red Sea over the last 220 000 years

Werner Ehrmann1, Paul A. Wilson2, Helge W. Arz3, Gerhard Schmiedl4

The manuscript is original and provides detailed information regarding the siliciclastic sediment compositions in core KL23 from the north part of the Red Sea. The data set provided in this manuscript is valuable and continues this group's work from recent years, where they infer paleoclimate trends from the isotopic values and clay minerals compositions. This contribution is important in providing high-resolution mineralogical and geochemical data and should be published for others to use. The discussion uses the available literature and draws broad spatial teleconnections based on various paleoclimate records and models. They provide a strong case for the ties between low latitude northern Red Sea and high latitudes ice caps glacial-interglacial cycle climate variability over the equatorial insolation driven variability seen southward in Red Sea archives.

The authors argue for a reasonable scenario – where during glacial periods and low global sea levels, the Nile River delta was exposed, and its sediments served as a significant source for terrigenous eolian sediments blown southward to the Red Sea. Their argument relies on increased smectite content and Ti counts during glacial periods, both likely originating from volcanic detritus. Another source of eolian sediments suggested by the authors to be significant in the past is the "Tokar Gap" and two other similar mountain gaps in the eastern borders of the Red Sea fringing mountain belt. These interpretations of the results might prove valid, however, other interpretations may well be inferred from the same results, following the discussion raised here:

**Line 155** – contrary to the stated argument the DSAF% maps from (Kunkelova et al., 2024) shows relatively high values for the region between Sallum (Egypt) and Benghazi (Libya). Indeed, this region is the source of reconstructed air parcel routes (Palchan and Torfstein, 2019). On the other hand, lower latitude East Sahara is not a probable source of dust to KL23 due to the local wind patterns and their convergence southward from it (e.g., Menezes et al., 2018).

Agreed, we have added the italicised clause to the sentence in question:

High contributions in the past from relatively inactive modern-day sources cannot be ruled out, but present-day DSAF maps (Schepanski et al., 2007; Kunkelova et al., 2022) strongly suggest that, today, *with the exception of a narrow strip of northeastern Cyreneaica (north easternmost Libya),* the northeastern corner of Africa (Egypt and eastern Libya) is a weak source of dust in comparison to other areas in the region such as the lower latitude Eastern Sahara, the Horn of Africa, the Levant, the central Arabian Peninsula, and Mesopotamia (see Kunkelova et al., 2024, their Fig. 9, for the most comprehensive DSAF data set).

Note that we now also include this coastal strip in an expanded discussion of potential unradiogenic dust sources to balance input to KL23 from the exposed Nile delta (lines 318-325).

**Line 180** – the treatment of removing marine barite is important but seems not very significant in interpretation of the provided Sr isotopes, as all of the previous terrigenous data from KL23 (Palchan

et al., 2013) is higher than the modern seawater composition of 0.709. Hence, as to the authors claim, it should be even more radiogenic than reported. Even so, comparing the Sr values in the current and previous work the difference seems to be negligible.

It is the size (not sign) of the offset in Sr isotope composition between barite (sea water) and the terrigenous fraction that determines the impact of barite contamination, but we agree that contamination is modest at KL23 compared to elsewhere, even within the Red Sea. Thus, we have added the following text to the first paragraph of Section 4.1:

Note that, while marine barite can severely contaminate terrigenous $^{87}Sr/^{86}Sr$, even where barite accumulation rates are modest (Jewell et al., 2022), that is not the case at KL23. This is because, here, the offset in Sr isotope composition between the terrigenous sediments supplied to the site (~0.7095 to ~0.7140) and barite (modern seawater) is modest in comparison to sites located elsewhere, for example, in the central Red Sea, the Mediterranean Sea and North Atlantic Ocean.

**Line 258 –** using the term "substantial" is a bit of a stretch as core KL23 smectite base levels are around 40% of the clay composition, thus, the rise during glacial periods is only additional 10%. This increase is proportional to the content of other clays as the analysis was done only on the <2um fraction. Thus, the rise could reflect decrease in other clays rather than more input from a specific source.

Furthermore, the concentration of the clay fraction in the samples drops significantly from ±12% to ±4% during the respective interval of increased smectite (Fig. 3B & Fig. 6C).
However, the use of εNd reflects sources without this issue and its low values "(typically ~ –8 to –6 εNd)" points to that if there is indeed a Deltaic source, it is surely not "substantial" as it resembles more granitoid detritus compared with the Deltaic higher εNd values.

Agreed, the term "substantial" is excessive and we replace it in the revised version with "distinct".

As a result of the closed sum effect, a reduction in the concentration of one clay mineral may be caused by an increase in the concentrations of other clay minerals. Smectite as the dominant clay mineral shows the largest amplitude in its concentration pattern. We consider it unlikely that the documented changes in smectite content were driven by dilution, because this explanation would require that the other four less abundant clay minerals would need to fluctuate together with one another. Thus, we infer that smectite variability is the primary pacemaker of change in the clay mineral assemblage.

We added a paragraph to the revised results section.

Methods remarks:

Section 2.3 – the leaching method is not specified. This is important and needs clarification and detail. Similarly, there is no detail on the analysis method (i.e., TIMS? Multi-collector?). Even if this is described in a previous paper, it is important to include minimal information regarding the method and analysis (indeed, this is discussed later in section 4.1). For example, what standard was used during the analysis, and what value was assumed for it?

We now include the following text in a more detailed methods in the supplements of the revised manuscript:

We treated our samples to remove all authigenic marine contaminants from the isotope fingerprint of the terrigenous fraction (carbonate, authigenic Fe-Mn oxyhydroxide coatings, organic carbon and marine barite) using the method of Jewell et al. (2022). That study demonstrated that small quantities of marine barite can have a large contaminating effect on the isotopic composition of the terrigenous fraction, especially for Sr. To remove marine barite, samples were leached with 0.2 M diethylene triamine pentaacetic acid (DTPA) in 2.5 M NaOH and left in a water bath set to 80 °C for 30 minutes. Following the recommended treatment protocol in Jewell et al. (2022), we repeated this step four times on three test samples to determine how many treatments were optimal for complete barite removal at this site. Marine sediments were considered free of marine barite when there was no appreciable decrease in Sr or Ba concentration, and no further change in measured $^{87}Sr / {}^{86}Sr$ of the silicate fraction with additional DTPA-NaOH treatment. Based on our test samples, we employed one DTPA-NaOH treatment (following the removal of all other marine phases). Analyses were performed by MC-ICPMS at the University of Southampton's Waterfront Campus.

Figure remarks:

**Fig. 2a** the window lacks a crucial potential source area depicted as increased DSAF% in northern Sahara around 20°N (Kunkelova et al., 2024). This region is a prominent source of air parcel reconstruction (Palchan and Torfstein, 2019).

We think the reviewer intended to say 30°N, but this is a good suggestion. We have expanded the window in Fig 2a as suggested. Furthermore, we modified the manuscript text in the first paragraph of Section 4 ("Discussion") and in Section 4.2 ("Sea-level controlled aeolian sediment influx from the Nile delta") to include the possibility of a dust contribution from the small coastal strip of Cyrenaica. We also now cite Palchan and Torfstein, (2019) on their air parcel analysis.

In summary, this is a fascinating paper with substantial data contribution on the clay mineralogical compositions in the northern Red Sea – a region largely overlooked. The conclusions drawn based on the results are partly debatable; the paleoclimate community will surely benefit from the discussion.

Thank you for the constructive comments on our contribution.

Daniel Palchan

References

Kunkelova T., Crocker A. J., Wilson P. A. and Schepanski K. (2024) Dust Source Activation Frequency in the Horn of Africa. *J. Geophys. Res. Atmospheres* **129**, e2023JD039694.

Menezes V. V., Farrar J. T. and Bower A. S. (2018) Westward mountain-gap wind jets of the northern Red Sea as seen by QuikSCAT. *Remote Sens. Environ.* **209**, 677–699.

Palchan D., Stein M., Almogi-Labin A., Erel Y. and Goldstein S. L. (2013) Dust transport and synoptic conditions over the Sahara–Arabia deserts during the MIS6/5 and 2/1 transitions from grain-size, chemical and isotopic properties of Red Sea cores. *Earth Planet. Sci. Lett.* **382**, 125–139.

Palchan D. and Torfstein A. (2019) A drop in Sahara dust fluxes records the northern limits of the African Humid Period. *Nat. Commun.* **10**.

---

## Author Comment (AC2)

**Review by Carlo Mologni**

Responses in red

The manuscript presents original research on siliciclastic sediment compositions in core KL23 from the northern Red Sea. The dataset is valuable as it extends the authors' previous work on paleoclimate trends through isotopic values and clay minerals. This study provides exceptionally high-resolution mineralogical and geochemical data supporting hypothesis on wind transport circulation between the Lower-Nile valley and the northern Read Sea over ~220 ka.

The discussion effectively integrates literature and establishes connections between northern Red Sea climate variability and glacial-interglacial cycles in high-latitude ice caps, contrasting with equatorial insolation-driven changes further south. The authors argue that during glacial periods and low sea levels, the exposed Nile River delta was a key source of eolian sediments, as indicated by increased smectite content, Ti counts and high εNd values.

However, some discrepancies exist between the data and the presented hypothesis. These discrepancies are not adequately explained, nor do the authors open the discussion to alternative hypotheses that deserve consideration. The following sections—GENERAL QUESTIONS, GENERAL COMMENTS, DETAILED COMMENTS, and FIGURE COMMENTS—highlight these issues.

Thank you for your positive assessment and especially for your useful and constructive comments and suggestions. We revised our manuscript accordingly.

**GENERAL QUESTIONS**

1) If, as hypothesized by the authors, the smectite fraction originates from the radiogenic Nile Delta sediments (average εNd ≈ -3) exposed during low sea level periods, why do high smectite and Ti concentrations during the Last Glacial Maximum (LGM) correspond to extremely low εNd (~ -8), which are characteristic of non-Nilotic sources, closer to the Saharan Shield?

From the perspective of North African dust sources, we do not consider -8 to be extremely low but we agree that it is low in comparison to the Ethiopian Highlands material carried north to the delta down the (Blue) Nile. Therefore, this is an important question. The key question is the extent to which the data sets available from the delta are representative of the real estate subaerially exposed by glacioeustatic sea level fall during glacial conditions. If we assume that the Nile delta sediments exposed during glacial conditions had a mean εNd composition as radiogenic as -3, the windblown dust supplied to KL23 must also have received a contribution from a more unradiogenic source. We invoked Lower Egypt, perhaps the Western Desert where dust sources, although relatively inactive today, are most unradiogenic. If on the other hand the downcore record (thanks for the suggestion) of Bastian et al. (2021) from the Nile fan, with εNd values down to almost -8, are representative of Nile delta deposits subaerially exposed by glacioeustatic sea level fall, the requirement for an additional unradiogenic dust source is lessened. Furthermore, Bastian et al. (2021) show that the silt fraction in their Nile fan core tends to carry a less radiogenic signal than the clay fraction. This could help to explain why our εNd data are less radiogenic than might be expected based on the high proportions of smectite in the clay fraction, because we measured εNd on the bulk terrigenous (not the clay) fraction.

We have overhauled the relevant paragraph in Section 4.2 to expand our discussion.

2) If smectite is associated with radiogenic Nile Delta sediments, as the authors suggest, why do low smectite values during the S5 period correspond to high εNd values (-1)? The authors interpret this period as one dominated by increased local sediment supply (chlorite). Does this imply that the northern Red Sea is also influenced by highly radiogenic local (non-aeolian) sources?

It is worth noting that the eastern margin of the northern Red Sea consists of recent (Oligocene to Quaternary) volcanic headwaters, which can serve as sources of smectite and high εNd radiogenic values (see:

*Antoine Delaunay, Guillaume Baby, Evelyn Garcia Paredes, Jakub Fedorik, Abdulkader M. Afifi, Evolution of the Eastern Red Sea Rifted Margin: Morphology, Uplift Processes, and Source-to-Sink Dynamics, Earth-Science Reviews, Volume 250, 2024, 104698, ISSN 0012-8252, https://doi.org/10.1016/j.earscirev.2024.104698).*

Agreed. These are aspects that we discussed in the original version, but this feedback suggests we were insufficiently clear. Therefore, we have overhauled the relevant discussion (see end of Section 4.4).

We now make more explicit in our discussion the distinction between the Arabian and African margins in the bedrock geology drained by their palaeoriver systems. We also go into more detail to discuss how our clay mineralogical and Nd isotope data sets are used to infer the provenance of the riverine fraction delivered to KL23 during AHP5 and AHP7.

3) If major and perennial fluvial sediment supply to KL23 is excluded, as proposed by the authors, the observed sedimentation rates appear disproportionately high compared to accumulation rates in the Nile Delta. This is particularly striking if KL23 sediments are assumed to be exclusively of aeolian origin.

How do the authors explain that aeolian sedimentation rates in the Red Sea are equal to or even higher than many fluvial sedimentation rates?

KL23 sediments cannot be assumed to be exclusively of aeolian origin, because they contain a large proportion of biogenic components (on average 63 %). Figure 3e shows that the terrigenous components in KL23 fluctuate between ca. 20 % and 50 %, with maxima occurring during glacial times and minima during interglacial periods. We have made this clearer in the manuscript Section 3.

Bulk sedimentation rates in the Red Sea are generally lower than in the Nile delta (we assume that the reviewer refers to core MS27PT from the Nile delta fan, Bastian et al., 2021; see below). Mean bulk sedimentation rates in KL23 in the northern Red Sea are 5.7 cm / kyr and they are ~5.2 cm / kyr in the central Red Sea at KL11 (Ehrmann et al., 2024). In contrast, bulk sedimentation rates in Core MS27PT (Bastian et al, 2021) are 6.6 cm / kyr. Sedimentation rates in the Nile delta are even higher (Blanchet et al., 2024).

These discrepancies remain unresolved. To address these issues, I encourage the authors to expand the discussion by considering additional hypotheses based on the data (see GENERAL COMMENTS below).

**GENERAL COMMENTS**

**A) The Gulf of Suez as a Sediment Source**

Based on source proxies (smectite and εNd ), the authors suggest that most of the KL23 sediment originates from Aeolian-reworked dust from the exposed Nile Delta during low sea level periods. However, none of the presented data directly confirm an eolian origin (e.g., grain surface analysis via exoscopy or grain-size distribution analysis).
The Gulf of Suez serves as a sediment repository for particles transported by marine currents from the Nile River. Therefore, high-smectite, radiogenic εNd sediments could simply originate from the erosion of the Gulf of Suez continental shelf during low sea level periods. The hypothesis that the Gulf

of Suez serves as a temporary, non-linear reservoir for high-smectite and radiogenic εNd sediments could provide a plausible explanation for Questions 1 and 2.

We are confused by this feedback, because we know of no marine connection between the Nile river and the Gulf of Suez. Thus, terrigenous sediment components in the Gulf of Suez have to be of aeolian origin. We discussed the potential of the Gulf of Suez as a sediment source in Section 4.2.

**B) The Role of Shallow and Deep-Water Circulation in the Red Sea**

The manuscript by Ehrmann et al. thoroughly discusses wind circulation around the study area, treating it as the main transport mechanism for clay particles at the KL23 site. However, it does not consider shallow or deep-water Red Sea circulation as a potential transport mode for smectites and radiogenic εNd sediments from the central/southern Red Sea.
As shown by Yao et al. (2014), shallow waters originating from the central/southern Red Sea reach the KL23 site (~25°N). These waters carry hydro-sedimentary inputs from the Eritrean/Ethiopian Basaltic Traps headwaters. Around 24°–25°N, sinking processes induce downwelling, potentially transporting sediment plumes rich in smectites and radiogenic εNd particles from the Barka River and other sources in Eritrea.

Please consider and develop this hypothesis in the discussion.

[Figure]

**Figure 14.** Schematic for the three-dimensional overturning circulation in the northern Red Sea. Most (0.5 Sv) of the surface western boundary current (0.6 Sv) crosses the basin at around 24°N, and then either sinks along the eastern boundary at the crossing latitude (0.1 Sv) or switches to an eastern boundary current (0.4 Sv) and sinks along the eastern boundary through a cyclonic recirculation. The downw-elled water at the intermediate depth is transported to the western boundary either through direct cross-basin flows or a rim current along the boundary. Meanwhile, the sinking along the eastern boundary is enhanced by a weaker cross-basin overturning circulation produced by the upwelling along the western boundary (0.2 Sv). A small portion of the western boundary current (0.1 Sv) sinks in the Gulf of Aqaba and Gulf of Suez and contributes to the intermediate and deep water.

Reference:
*Yao, F., Hoteit, I., Pratt, L. J., Bower, A. S., Kohl, A., Gopalakrishnan, G., & Rivas, D. (2014). Seasonal Overturning Circulation in the Red Sea: 2. Winter Circulation. J. Geophys. Res. Oceans, 119, 2263–2289. doi:10.1002/2013JC009331.*

In Ehrmann et al. (2024) we showed that the main source of fluvial input to the central Red Sea during AHPs was the Baraka fluvial system, but the composition of this material is not dominated by volcanic debris such as smectite and Ti (see below, response to "Detailed comment Line 216"). Instead, influx of smectite and Ti into the central Red Sea is chiefly by aeolian transport through Tokar Gap and that process is strongly paced by precession (Fig. 4; Ehrmann et al., 2024). Thus, we can rule out the possibility that transportation of this material to the northern Red Sea by oceanic currents was a major process because their accumulation at KL23 shows a glacial-interglacial timescale pacing of change. Furthermore, Ti abundance along a N-S transect in the northern Red Sea decreases from N to S (Fig.

S5) and smectite concentrations in core KL23 are at times higher than in core KL11 in the central Red Sea, both implying a source in the north.

We have added text to Section 4.2 of the revised manuscript to make this clear.

**C) Sedimentation Rates and εNd Variability between the Nile Delta and KL23**

Sedimentation rates at KL23 are notably high compared to those in the Nile Delta. Similarly, the average εNd values often overlap with those from Nile Delta coring sites. Additional data supporting and discussing source correlations with the study site would be beneficial. For example, a 100-ka-long dataset of εNd , smectite, and sedimentation rates from the Nile Deep Delta Fan is available in: *Luc Bastian & Carlo Mologni, Nathalie Vigier, Germain Bayon, Henry Lamb, Delphine Bosch, Marie-Emmanuelle Kerros, Christophe Colin, Marie Revel, Co-variations of Climate and Silicate Weathering in the Nile Basin during the Late Pleistocene, Quaternary Science Reviews, Volume 264, 2021, 107012, ISSN 0277-3791, https://doi.org/10.1016/j.quascirev.2021.107012*

Bulk sedimentation rates in the Red Sea are generally lower than in the Nile delta fan and the Nile delta! Mean bulk sedimentation rates in KL23 in the northern Red Sea are 5.7 cm /k yr, in the central Red Sea core they are 5.2 cm / kyr (Ehrmann et al., 2024). In contrast, bulk sedimentation rates in Core MS27PT (Bastian et al, 2021) are 6.6 cm / kyr. Sedimentation rates in the Nile delta are even higher (Blanchet et al., 2024). Taking into account that KL23 has a mean carbonate content of 63 % (biogenic pelagic components) and the concentration of the components of terrigenous origin is only 20 % to 50 % (Fig. 3), the influx rates of terrigenous matter are much lower.

Sedimentation rates and smectite abundances in the Nile delta, in the Nile delta fan and under the Nile discharge plume in the Eastern Mediterranean Sea are mainly controlled by sediment discharge through the Nile, and thus by the climate in tropical Africa. Most of the Nile sediment, including smectite, comes from the Ethiopian Highlands. Therefore, the proportions of the clays fluctuate according to the intensity of the precession cycle (e.g., Revel et al., 2010; Ehrmann et al., 2016; Bastian et al., 2021). The Nile sediment discharge, however, has no direct influence on the aeolian sediment transport from the Nile delta to the northern Red Sea, which is controlled by the size of the desiccated delta. Therefore, a correlation of Nile delta sedimentation and sedimentation at KL23 in the northern Red Sea, in our opinion is dispensable.

We have clarified the corresponding text of Section 4.2: "changing Nile sediment discharge rates that fluctuate in accordance with the precession cycles (e.g., Revel et al., 2010; Ehrmann et al., 2016; Bastian et al., 2021) have no direct influence on the aeolian sediment transport from the Nile delta to the northern Red Sea, because the latter process is controlled by the size of the exposed delta area".

We will also refer to the Nd data of Bastian et al. (2021) in Section 4.2.

For our response on Nd isotope composition we refer the reviewer to our reply to GENERAL COMMENT 1 (above).

- - - - - - - - - - -

**DETAILED COMMENTS**

**Line 26**: Specify which grain size fraction is being analyzed.

We refer to the clay fraction of the sediments. Specified during revision.

**Line 80**: Provide a synthetic figure of the age model as a review of all ages given in previous works. The manuscript also needs a presentation of sedimentation rates in the sequence.

We now present an age / depth plot with calculated bulk sedimentation rates for sediment core KL23 in the supplements.

**Line 91**: Include a detailed protocol in the supplementary materials.

Although we gave a short description of the methods including references to papers that describe the methods in detail, we now follow the advice of the reviewer and present a more comprehensive description of the methods in the supplements.

**Line 102**: Explain why these elements are representative of the terrigenous sediment fraction, with mineralogical justification or bibliographic support.

Al, Si, K, Ti, Rb, and Zr commonly occur in rocks like granitoid, metamorphic and volcanic rocks and clastic sediments. Rocks of these types and their weathering products surround the northern Red Sea. The elements are common components of minerals such as quartz, feldspar, mica, amphibole, pyroxene, zircon, rutile (e.g., Croudace and Rothwell, 2015). We now add a sentence to the methods chapter.

**Line 141**: "Modest maxima of up to 20%"—20% of what? Please specify.

We now specify this refers to sand in the revised manuscript.

**Line 215**: There are multiple volcanic sources on the Arabian headwaters of the Red Sea; consider citing: *Antoine Delaunay, Guillaume Baby, Evelyn Garcia Paredes, Jakub Fedorik, Abdulkader M. Afifi, Evolution of the Eastern Red Sea Rifted margin: morphology, uplift processes and source-to-sink dynamics, Earth-Science Reviews, Volume 250, 2024, 104698, ISSN 0012-8252, https://doi.org/10.1016/j.earscirev.2024.104698*.

We have incorporated this reference. Thanks for the suggestion.

**Line 216**: The Barka River, originating in the Eritrean Highlands over a basaltic trap context, flows northward into the Red Sea (~640 km in length). Given the northward shallow water circulation in the Red Sea (see GENERAL COMMENT B), its potential recent volcanic sediment contributions to the northern Red Sea should be considered.

We are well aware of the seasonal Baraka (Barka) river and its main tributaries Anseba river and Langeb river. The catchment covers some 66 000 km$^2$. The Baraka system discharges through the Tokar delta into the central Red Sea. It is active for 40–70 days per year, mainly during autumn, with an annual water discharge of 200–970 x 10$^6$ m$^3$ (Trommer et al., 2011). No information is available to us about the amount of sediment discharge. The headwaters in the Red Sea Hills and Eritrean Highland are composed of rocks of the Arabian Nubian Shield, mainly Precambrian gneisses, schists and granitoids (see geological maps of Sudan and Eritrea: GMRD, 1981, https://esdac.jrc.ec.europa.eu/content/geological-map-sudan / Abbate, E., Billi, P.: Geology and Geomorphological Landscapes of Eritrea, In: Billi, P. (Ed.) Landscapes and Landforms of the Horn of Africa. World Geomorphological Landscapes. Springer, Cham. https://doi.org/10.1007/978-3-031-05487-7_2, 2022). The trap basalts mentioned by the reviewer form a N–S elongated strip and cover only ca. 2500 km$^2$. The Baraka river has its source in the northernmost corner, near the town Asmara, and flows only a few kilometres through the basalts, which limits the uptake of volcanic debris.

We analysed sediments from a sediment core, KL11, in the central Red Sea off Tokar delta (Ehrmann et al., 2024). Sediments of the Eemian humid period, when the Baraka drainage system presumably was especially active and aeolian influx into the Red Sea was minor, are poor in smectite and show a low Ti / terr ratio. Also, the radiogenic isotope composition of the terrigenous fraction in KL11 recorded for the Eemian humid period is consistent with a strong imprint of fluvial input, but not of volcanic debris.

Furthermore, fluvial discharge into the central and southern Red Sea is controlled by the monsoonal rain belt and follows the precession cycle. In the northern Red Sea core KL23 the abundance of the volcanic-derived components smectite and Ti, however, follows strongly the glacial/interglacial cycles. Thus, an oceanic transport of suspension does not play a major role.

No changes to the manuscript undertaken.

**Line 379**: Specify what "fine grain size" refers to.

We refer to the clay fraction of the sediments here. Specified in our revision.

- - - - - - - - - - -

**FIGURE COMMENTS**

**Figure 1a**: Please add εNd values to the map, corresponding to geological regions in Africa and the Arabian Peninsula.

We refrain from including εNd data in Fig. 1a because of lucidity and because such data are presented in Fig. 2b.

**Figures 4 & 5**: Add precession and eccentricity curves to support hypotheses discussed in the main text. Correlating εNd, smectite, and sedimentation rates with Nile Delta sequences is strongly recommended.

The influence of eccentricity and precession on sedimentation in the central and northern Red Sea is seen in Fig. 4 (right panels), but we now include the eccentricity and precession index to Figs. 3, 5 and 6 in the revised manuscript.

- - - - - - - - - - -

This review would enhance the text's clarity and introduces additional plausible hypotheses that should be considered in this article.

Carlo Mologni